# FEDERATED CAUSAL DISCOVERY FROM HETEROGENEOUS DATA

**Loka Li**[1], **Ignavier Ng**[2], **Gongxu Luo**[1], **Biwei Huang**[3],
**Guangyi Chen**[1,2], **Tongliang Liu**[1], **Bin Gu**[1], **Kun Zhang**[1,2]
[1] Mohamed bin Zayed University of Artificial Intelligence
[2] Carnegie Mellon University
[3] University of California San Diego
{longkang.li, kun.zhang}@mbzuai.ac.ae

## ABSTRACT

Conventional causal discovery methods rely on centralized data, which is inconsistent with the decentralized nature of data in many real-world situations. This discrepancy has motivated the development of federated causal discovery (FCD) approaches. However, existing FCD methods may be limited by their potentially restrictive assumptions of identifiable functional causal models or homogeneous data distributions, narrowing their applicability in diverse scenarios. In this paper, we propose a novel FCD method attempting to accommodate arbitrary causal models and heterogeneous data. We first utilize a surrogate variable corresponding to the client index to account for the data heterogeneity across different clients. We then develop a federated conditional independence test (FCIT) for causal skeleton discovery and establish a federated independent change principle (FICP) to determine causal directions. These approaches involve constructing summary statistics as a proxy of the raw data to protect data privacy. Owing to the nonparametric properties, FCIT and FICP make no assumption about particular functional forms, thereby facilitating the handling of arbitrary causal models. We conduct extensive experiments on synthetic and real datasets to show the efficacy of our method. The code is available at `https://github.com/lokali/FedCDH.git`.

## 1 INTRODUCTION

Causal discovery aims to learn the causal structure from observational data, attracting significant attention from fields such as machine learning and artificial intelligence (Nogueira et al., 2021), healthcare (Shen et al., 2020), economics (Zhang & Chan, 2006), manufacturing (Vuković & Thalmann, 2022) and neuroscience (Tu et al., 2019). Recently, it has been facing new opportunities and challenges from the rapid growth of data volume. One of the key challenges is data decentralization. Traditionally, causal discovery is conducted at a centralized site where all data is gathered in one location. However, in real-world scenarios, data is often distributed across multiple parties, such as the healthcare data across various hospitals (Kidd et al., 2022). Consequently, there has been increasing interest in federated causal discovery (FCD), which aims to uncover the underlying causal structure of decentralized data with privacy and security concerns.

Existing FCD methods from observational data can be generally classified as continuous-optimization-based, constraint-based, and score-based methods. Some continuous-optimization-based methods extend NOTEARS (Zheng et al., 2018) with federated strategies, such as NOTEARS-ADMM (Ng & Zhang, 2022) that relies on the ADMM (Boyd et al., 2011) optimization method, FedDAG (Gao et al., 2022) that employs FedAvg (McMahan et al., 2017) technique, and FED-CD (Abyaneh et al., 2022) that utilizes belief aggregation. These methods might suffer from various technical issues, including convergence (Wei et al., 2020; Ng et al., 2022), nonconvexity (Ng et al., 2023), and sensitivity to data standardization (Reisach et al., 2021). As for score-based methods, DARLS (Ye et al., 2022) utilizes the distributed annealing (Arshad & Silaghi, 2004) strategy to search for the optimal graph, while PERI (Mian et al., 2023) aggregates the results of the local greedy equivalent search (GES) (Chickering, 2002) and chooses the worst-case regret for each iteration. One constraint-based method, FED$C^2$SL (Wang et al., 2023), extends $\chi^2$ test to the federated

Table 1: The comparison of related works about federated causal discovery from observational data. Our FedCDH method could handle arbitrary functional causal models and heterogeneous data.

| Method | Category | Data Distribution | Causal Model Assumption [1] | Identifiability Requirement | Federated Strategy |
|--------|----------|-------------------|------------------------------|------------------------------|---------------------|
| NOTEARS-ADMM (Ng & Zhang, 2022) | Optimization-based | Homogeneous | Linear Gaussian, EV | Yes | ADMM |
| NOTEARS-MLP-ADMM (Ng & Zhang, 2022) | Optimization-based | Homogeneous | Nonlinear Additive Noise | Yes | ADMM |
| FedDAG (Gao et al., 2022) | Optimization-based | Heterogeneous | Nonlinear Additive Noise | Yes | FedAvg |
| Fed-CD (Abyaneh et al., 2022) | Optimization-based | Heterogeneous | Nonlinear Additive Noise | Yes | Belief Aggregation |
| DARLIS (Ye et al., 2022) | Score-based | Homogeneous | Generalized Linear | No | Distributed Annealing |
| PERI (Mian et al., 2023) | Score-based | Homogeneous | Linear Gaussian, NV | No | Voting |
| FedPC (Huang et al., 2022) | Constraint-based | Homogeneous | Linear Gaussian, NV | No | Voting |
| FedCDH (Ours) | Constraint-based | Heterogeneous | Arbitrary Functions | No | Summary Statistics |

version, however, this method is restrictive on discrete variables and therefore not applicable for any continuous variables. Other constraint-based methods, such as FedPC (Huang et al., 2022), aggregate the skeletons and directions of the Peter-Clark (PC) algorithm (Spirtes et al., 2000) by each client via a voting mechanism. However, as shown in Table 1, most of these methods heavily rely on either identifiable functional causal models or homogeneous data distributions. These assumptions may be overly restrictive and difficult to be satisfied in real-world scenarios, limiting their diverse applicability. For instance, distribution shifts may often occur in the real world across different clients owing to different interventions, collection conditions, or domains, resulting in the presence of heterogeneous data. Please refer to Appendix A2 for further discussion of related works, including those of causal discovery, heterogeneous data and FCD.

In this paper, we propose FedCDH, a novel constraint-based approach for **Fed**erated **C**ausal **D**iscovery from **H**eterogeneous data. The primary innovation of FedCDH lies in using summary statistics as a proxy for raw data during skeleton discovery and direction determination in a federated fashion. Specifically, to address heterogeneous data, we first introduce a surrogate variable corresponding to the client or domain index, allowing our method to model distribution changes. Unlike existing FCD methods that only leverage the data from different clients to increase the total sample size, we demonstrate how such data heterogeneity across different clients benefits the identification of causal directions and how to exploit it. Furthermore, we propose a federated conditional independence test (FCIT) for causal skeleton discovery, incorporating random features (Rahimi & Recht, 2007) to approximate the kernel matrix which facilitates the construction of the covariance matrix. Additionally, we develop a federated independent change principle (FICP) to determine causal directions, exploiting the causal asymmetry. FICP also employs random features to approximate the embeddings of heterogeneous conditional distributions for representing changing causal models. It is important to note that FCIT and FICP are non-parametric, making no assumption about specific functional forms, thus facilitating the handling of arbitrary causal models. To evaluate our method, we conduct extensive experiments on synthetic datasets including linear Gaussian models and general functional models, and real-world dataset including fMRI Hippocampus (Poldrack et al., 2015) and HK Stock Market datasets (Huang et al., 2020). The significant performance improvements over other FCD methods demonstrate the superiority of our approach.

## 2 REVISITING CAUSAL DISCOVERY FROM HETEROGENEOUS DATA

In this section, we will firstly provide an overview of causal discovery and some common assumptions, then we will introduce the characterizations of conditional independence and independent change. This paper aims at extending those techniques from the centralized to the federated setting.

**1) Causal Discovery with Changing Causal Models.**

Consider $d$ observable random variables denoted by $\boldsymbol{V} = (V_1, \ldots, V_d)$ and $K$ clients, and one client corresponds to one unique domain. In this paper, we focus on horizontally-partitioned data (Samet & Miri, 2009), where each client holds a different subset of the total data samples while all the clients share the same set of features. Let $k$ be the client index, and $\eth$ be the domain index, where $k, \eth \in \{1, \ldots, K\}$. Each client has $n_k$ samples, in total there are $n$ samples, denoted by

---
[1]Linear Gaussian model with equal noise variance (EV) (Peters & Bühlmann, 2014) and nonlinear additive noise model (Hoyer et al., 2008) are identifiable, while linear Gaussian model with non-equal noise variance (NV) is not identifiable. Generalized linear model and arbitrary functional model are certainly not identifiable.

$n = \sum_{k=1}^{K} n_k$. The task of federated causal discovery is to recover the causal graph $\mathcal{G}$ given the decentralized data matrix $\boldsymbol{V} \in \mathbb{R}^{n \times d}$.

When the data is homogeneous, the causal process for each variable $V_i$ can be represented by the following structural causal model (SCM): $V_i = f_i(\text{PA}_i, \epsilon_i)$, where $f_i$ is the causal function, $\text{PA}_i$ is the parents of $V_i$, $\epsilon_i$ is a noise term with non-zero variance, and we assume the $\epsilon_i$'s are mutually independent. When the data is heterogeneous, there must be some causal models changing across different domains. The changes may be caused by the variation of causal strengths or noise variances. Therefore, we formulate the causal process for heterogeneous data as: $V_i = f_i(\text{PA}_i, \epsilon_i, \theta_i(\mho), \tilde{\psi}(\mho))$, where $\mho$ is regarded as an observed random variable referred as the domain index, the function $f_i$ or the distribution of the noise $\epsilon_i$ is different or changing across different domains, both $\tilde{\psi}(\mho)$ and $\theta_i(\mho)$ are unobserved domain-changing factors represented as the functions of variable $\mho$, $\tilde{\psi}(\mho)$ is the set of "pseudo confounders" that influence the whole set of variables and we assume there are $L$ such confounders ($\tilde{\psi}(\mho) = \{\psi_l(\mho)\}_{l=1}^{L}$, the minimum value for $L$ can be 0 meaning that there is no such latent confounder in the graph, while the maximum value can be $\mathbb{C}_d^2 = \frac{d(d+1)}{2}$, meaning that each pair of observed variables has a hidden confounder), $\theta_i(\mho)$ denotes the effective parameters of $V_i$ in the model, and we assume that $\theta_i(\mho)$ is specific to $V_i$ and is independent of $\theta_j(\mho)$ for any $i \neq j$. $\tilde{\psi}(\mho)$ and $\theta_i(\mho)$ input $\mho$ which is a positive integer and output a real number. Let $\mathcal{G}_{obs}$ be the underlying causal graph over $\boldsymbol{V}$, and $\mathcal{G}_{aug}$ be the augmented graph over $\boldsymbol{V} \cup \tilde{\psi}(\mho) \cup \{\theta_i(\mho)\}_{i=1}^{d}$. For causal discovery with changing causal models, we follow previous work such as CD-NOD (Huang et al., 2020) and make the following assumptions.

**Assumption 1** (Pseudo Causal Sufficiency). *There is no confounder in the dataset of one domain, but we allow the changes of different causal modules across different domains to be dependent.*

**Assumption 2** (Markov and Faithfulness). *The joint distribution over $\boldsymbol{V} \cup \tilde{\psi}(\mho) \cup \{\theta_i(\mho)\}_{i=1}^{d}$ is Markov and faithful to $\mathcal{G}_{aug}$.*

To remove the potential influence from confounders and recover causal relations across different domains, causal discovery could be conducted on the augmented graph $\mathcal{G}_{aug}$ instead of $\mathcal{G}_{obs}$. While $\tilde{\psi}(\mho) \cup \{\theta_i(\mho)\}_{i=1}^{d}$ are unobserved variables, the domain index $\mho$ is observed variable. Therefore, $\mho$ is introduced as the surrogate variable (Huang et al., 2020) for causal discovery from heterogeneous data. An illustration is given in Figure 1, where the augmented graph with the unobserved domain-changing variables $\tilde{\psi}(\mho)$ and $\theta_i(\mho)$ could be simplified by an augmented graph with just a surrogate variable $\mho$. If there is an edge between surrogate variable $\mho$ and observed variable $V_i$ on $\mathcal{G}_{aug}$, then it means that the causal model related to $V_i$ is changing across different domains, in other words, the data distribution of $V_i$ is heterogeneous across domains.

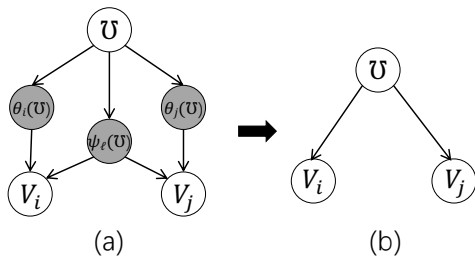

Figure 1: An illustration where the causal models of variables $V_i$ and $V_j$ are changing across domains. (a) the graph with unobserved domain-changing factors $\psi_\ell(\mho)$, $\theta_i(\mho)$ and $\theta_j(\mho)$; (b) the simplified graph with the surrogate variable $\mho$.

**2) Characterization of Conditional Independence.** Let $X, Y, Z$ be random variables or sets of random variables, with the domains $\mathcal{X}, \mathcal{Y}, \mathcal{Z}$, respectively. Define a measurable and positive definite kernel $k_{\mathcal{X}}$, and denote the corresponding reproducing kernel Hilbert space (RKHS) $\mathcal{H}_{\mathcal{X}}$. Similarly, we define $k_{\mathcal{Y}}, \mathcal{H}_{\mathcal{Y}}, k_{\mathcal{Z}}$ and $\mathcal{H}_{\mathcal{Z}}$. One of the most used characterizations of conditional independence (CI) is: $X \perp\!\!\!\perp Y | Z$ if and only if $\mathbb{P}_{XY|Z} = \mathbb{P}_{X|Z} \mathbb{P}_{Y|Z}$, or equivalently $\mathbb{P}_{X|Y,Z} = \mathbb{P}_{X|Z}$. Another characterization of CI is given in terms of the partial cross-covariance operator on RKHS.

**Lemma 1** (Characterization of CI with Partial Cross-covariance (Fukumizu et al., 2007)). *Let $\ddot{X} \triangleq (X, Z), k_{\ddot{X}} \triangleq k_{\mathcal{X}} k_{\mathcal{Z}}$, and $\mathcal{H}_{\ddot{X}}$ be the RKHS corresponding to $k_{\ddot{X}}$. Assume that $\mathcal{H}_{\mathcal{X}} \subset L_X^2, \mathcal{H}_{\mathcal{Y}} \subset L_Y^2, \mathcal{H}_{\mathcal{Z}} \subset L_Z^2$. Further assume that $k_{\ddot{X}} k_{\mathcal{Y}}$ is a characteristic kernel on $(\mathcal{X} \times \mathcal{Z}) \times \mathcal{Y}$, and that $\mathcal{H}_{\mathcal{Z}} + \mathbb{R}$ (the direct sum of two RHKSs) is dense in $L^2(\mathbb{P}_{\mathbb{Z}})$. Let $\Sigma_{\ddot{X}Y|Z}$ be the partial cross-covariance operator, then*

$$\Sigma_{\ddot{X}Y|Z} = 0 \iff X \perp\!\!\!\perp Y | Z. \tag{1}$$

Based on the above lemma, we further consider a different characterization of CI which enforces the uncorrelatedness of functions in suitable spaces, which may be intuitively more appealing. More details about the interpretation of $\Sigma_{\ddot{X}Y|Z}$, the definition of characteristic kernel, and the uncorrelatedness-based characterization of CI, are put in Appendix A3.1.

**3) Characterization of Independent Change.** The Hilbert-Schmidt Independence Criterion (HSIC) (Gretton et al., 2007) is a statistical measure used to assess the independence between two random variables in the RKHS. We use the normalized HSIC to evaluate the independence of two changing causal models. The value of the normalized HSIC ranges from 0 to 1, and a smaller value indicates that the two changing causal models are more independent. Let $\hat{\triangle}_{X \to Y}$ be the normalized HSIC between $\mathbb{P}(X)$ and $\mathbb{P}(Y|X)$, and $\hat{\triangle}_{Y \to X}$ be the normalized HSIC between $\mathbb{P}(Y)$ and $\mathbb{P}(X|Y)$. Then, we can determine the causal direction between $X$ and $Y$ with the following lemma.

**Lemma 2** (Independent Change Principle (Huang et al., 2020)). *Let $X$ and $Y$ be two random observed variables. Assume that both $X$ and $Y$ have changing causal models (both of them are adjacent to $\mho$ in $\mathcal{G}_{aug}$). Then the causal direction between $X$ and $Y$ can be determined according to the following rules*

    *i) If $\hat{\triangle}_{X \to Y} < \hat{\triangle}_{Y \to X}$, output the direction $X \to Y$,*

    *ii) If $\hat{\triangle}_{X \to Y} > \hat{\triangle}_{Y \to X}$, output the direction $Y \to X$.*

More details about the definition and formulation of $\hat{\triangle}_{X \to Y}$ and $\hat{\triangle}_{X \to Y}$ are in Appendix A3.2. It is important to note that: once the Gaussian kernel is utilized, the kernel-based conditional independence test (Zhang et al., 2012) and the kernel-based independent change principal (Huang et al., 2020) assume smoothness for the relationship of continuous variables.

# 3    FEDERATED CAUSAL DISCOVERY FROM HETEROGENEOUS DATA

In this section, we will explain our proposed FedCDH method in details. An overall framework of FedCDH is given in Figure 2. Two key submodules of our method are federated conditional independent test (FCIT; Theorem 4 and Theorem 5) and federated independent change principle (FICP; Theorem 6), which are presented in Section 3.1 and Section 3.2, respectively. We then illustrate how to construct the summary statistics and how to implement FCIT and FICP with summary statistics (Theorem 8) in Section 3.3. Last but not least, we discuss the communication costs and secure computations in Section 3.4. For the proofs of theorems and lemmas, please refer to Appendix A4.

## 3.1    FEDERATED CONDITIONAL INDEPENDENT TEST (FCIT)

**1) Null Hypothesis.** Consider the null and alternative hypotheses

$$\mathcal{H}_0 : X \perp\!\!\!\perp Y|Z, \quad \mathcal{H}_1 : X \not\perp\!\!\!\perp Y|Z. \tag{2}$$

According to Eq. 1, we can measure conditional independence based on the RKHSs. Therefore, we equivalently rewrite the above hypothesis more explicitly as

$$\mathcal{H}_0 : \|\Sigma_{\ddot{X}Y|Z}\|_{HS}^2 = 0, \quad \mathcal{H}_1 : \|\Sigma_{\ddot{X}Y|Z}\|_{HS}^2 > 0. \tag{3}$$

Note that the computation forms of Hilbert-Schmidt norm and Frobenius norm are the same, and the difference is that the Hilbert-Schmidt norm is defined in infinite Hilbert space while the Frobenius norm is defined in finite Euclidean space. We here consider the squared Frobenius norm of the empirical partial cross-covariance matrix as an approximation for the hypotheses, given as

$$\mathcal{H}_0 : \|\mathcal{C}_{\ddot{X}Y|Z}\|_F^2 = 0, \quad \mathcal{H}_1 : \|\mathcal{C}_{\ddot{X}Y|Z}\|_F^2 > 0, \tag{4}$$

where $\mathcal{C}_{\ddot{X}Y|Z} = \frac{1}{n} \sum_{i=1}^{n} [(\ddot{A}_i - \mathbb{E}(\ddot{A}|Z))^T (B_i - \mathbb{E}(B|Z))]$ corresponds to the partial cross-covariance matrix with $n$ samples, $\mathcal{C}_{\ddot{X}Y|Z} \in \mathbb{R}^{h \times h}$, $\ddot{A} = f(\ddot{X}) \in \mathbb{R}^{n \times h}$, $B = g(Y) \in \mathbb{R}^{n \times h}$, $\{f^j(\ddot{X})|_{j=1}^h\} \in \mathcal{F}_{\ddot{X}}$, $\{g^j(Y)|_{j=1}^h\} \in \mathcal{F}_Y$. $n$ and $h$ denote the number of total samples of all clients and the number

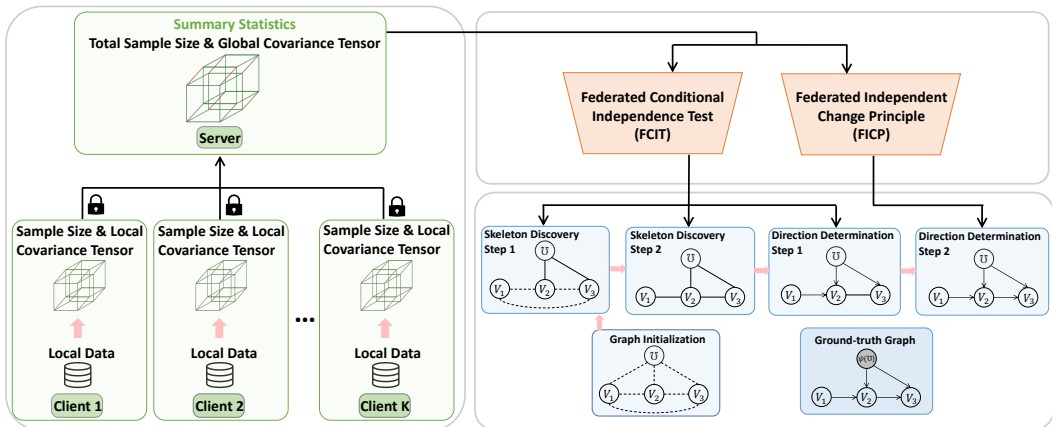

Figure 2: Overall framework of FedCDH. *Left*: The clients will send their sample sizes and local covariance tensors to the server, for constructing the summary statistics. The federated causal discovery will be implemented on the server. *Right Top*: Relying on the summary statistics, we propose two submodules: federated conditional independence test and federated independent change principle, for skeleton discovery and direction determination. *Right Bottom*: An example of FCD with three observed variables is illustrated, where the causal modules related to $V_2$ and $V_3$ are changing.

of hidden features or mapping functions, respectively. Since $\ddot{X} \triangleq (X, Z)$, then for each function $f^j : \ddot{X} \mapsto \mathcal{F}_{\ddot{X}}$, the input is $\ddot{X} \in \mathbb{R}^{n \times 2}$ and the output $f^j(\ddot{X}) \in \mathbb{R}^n$. For each function $g^k : Y \mapsto \mathcal{F}_Y$, the input is $Y \in \mathbb{R}^n$ and the output $g^k(Y) \in \mathbb{R}^n$. Notice that $\mathcal{F}_{\ddot{X}}$ and $\mathcal{F}_Y$ are function spaces, which are set to be the support of the process $\sqrt{2} \cos(\boldsymbol{w} \cdot + \boldsymbol{b})$, $\boldsymbol{w}$ follows standard Gaussian distribution, and $\boldsymbol{b}$ follows uniform distribution from $[0, 2\pi]$. We choose these specific spaces because in this paper we use random features to approximate the kernels. $\mathbb{E}(\ddot{A}|Z)$ and $\mathbb{E}(B|Z)$ could be non-linear functions of $Z$ which are difficult to estimate. Therefore, we would like to approximate them with linear functions. Let $q(Z) \in \mathbb{R}^{n \times h}$, $\{q^j(Z)|_{j=1}^h\} \in \mathcal{F}_Z$, $\mathcal{F}_Z$ shares a similar function space with $\mathcal{F}_Y$. We could estimate $\mathbb{E}(f^j|Z)$ with the linear ridge regression solution $u_j^T q(Z)$ and estimate $\mathbb{E}(g^j|Z)$ with $v_j^T q(Z)$ under mild conditions (Sutherland & Schneider, 2015). Now we give the following lemma.

**Lemma 3** (Characterization of Conditional Independence). *Let $f^j$ and $g^j$ be the functions defined for the variables $\ddot{X}$ and $Y$, respectively. Then $X \perp\!\!\!\perp Y | Z$ is approximated by the following condition*

$$\mathbb{E}(\tilde{f}\tilde{g}) = 0, \ \forall \tilde{f} \in \mathcal{F}_{\ddot{X}|Z} \text{ and } \forall \tilde{g} \in \mathcal{F}_{Y|Z}, \tag{5}$$

*where $\mathcal{F}_{\ddot{X}|Z} = \{\tilde{f} \mid \tilde{f}^j = f^j - u_j^T q(Z), \ f^j \in \mathcal{F}_{\ddot{X}}\}$ and $\mathcal{F}_{Y|Z} = \{\tilde{g} \mid \tilde{g}^j = g^j - v_j^T q(Z), \ g^j \in \mathcal{F}_Y\}$.*

Let $\gamma$ be a small ridge parameter. According to Eq. 1 and Eq. 5, by ridge regression, we obtain

$$\mathcal{C}_{\ddot{X}Y|Z} = \mathcal{C}_{\ddot{X}Y} - \mathcal{C}_{\ddot{X}Z}(\mathcal{C}_{ZZ} + \gamma \boldsymbol{I})^{-1}\mathcal{C}_{ZY}. \tag{6}$$

**2) Test Statistic and Null Distribution.** In order to ensure the convergence to a non-degenerate distribution, we multiply the empirical estimate of the Frobenius norm by $n$, and set it as the test statistic $\mathcal{T}_{CI} = n\|\mathcal{C}_{\ddot{X}Y|Z}\|_F^2$. Let $\tilde{K}_{\ddot{X}|Z}$ be the centralized kernel matrix, given by $\tilde{K}_{\ddot{X}|Z} \triangleq \boldsymbol{H}R_{\ddot{X}|Z}R_{\ddot{X}|Z}^T\boldsymbol{H}$, where $\boldsymbol{H} = \boldsymbol{I} - \frac{1}{n}\boldsymbol{1}\boldsymbol{1}^T$ and $R_{\ddot{X}|Z} \triangleq \tilde{f}(\ddot{X}) = f(\ddot{X}) - u^T q(Z)$ which can be seen as the residual after kernel ridge regression. Here, $\boldsymbol{I}$ refers to the $n \times n$ identity matrix and $\boldsymbol{1}$ denotes the vector of $n$ ones. We now define $\tilde{K}_{Y|Z}$ similarly. Let $\lambda_{\ddot{X}|Z}$ and $\lambda_{Y|Z}$ be the eigenvalues of $\tilde{K}_{\ddot{X}|Z}$ and $\tilde{K}_{Y|Z}$, respectively. Let $\{\alpha_1, \ldots, \alpha_L\}$ denote i.i.d. standard Gaussian variables, and thus $\{\alpha_1^2, \ldots, \alpha_L^2\}$ denote i.i.d. $\chi_1^2$ variables. Considering $n$ i.i.d. samples from the joint distribution $\mathbb{P}_{\ddot{X}YZ}$, we have

**Theorem 4** (Federated Conditional Independent Test). *Under the null hypothesis $\mathcal{H}_0$ ($X$ and $Y$ are conditionally independent given $Z$), the test statistic*

$$\mathcal{T}_{CI} \triangleq n\|\mathcal{C}_{\ddot{X}Y|Z}\|_F^2, \tag{7}$$

*has the asymptotic distribution*

$$\hat{\mathcal{T}}_{CI} \triangleq \frac{1}{n^2} \sum_{i,j=1}^{L} \lambda_{\ddot{X}|Z,i} \lambda_{Y|Z,j} \alpha_{ij}^2.$$

Although the defined test statistics are equivalent to that of kernel-based conditional independence test (KCI) (Zhang et al., 2012), the asymptotic distributions are in different forms. Please note that the large sample properties are needed when deriving the asymptotic distribution $\hat{\mathcal{T}}_{CI}$ above, and the proof is shown in Appendix A4.2.

Given that $X \perp\!\!\!\perp Y|Z$, we could introduce the independence between $R_{\ddot{X}|Z}$ and $R_{Y|Z}$, which leads to the separation between $\lambda_{\ddot{X}|Z,i}$ and $\lambda_{Y|Z,j}$. We show that this separated form could help to approximate the null distribution in terms of a decomposable statistic, such as the covariance matrix.

We approximate the null distribution with a two-parameter Gamma distribution, which is related to the mean and variance. Under the hypothesis $\mathcal{H}_0$ and given the sample $\mathcal{D}$, the distribution of $\hat{\mathcal{T}}_{CI}$ can be approximated by the $\Gamma(\hat{k}, \hat{\theta})$ distribution: $\mathbb{P}(t) = (t^{\hat{k}-1} \cdot e^{-t/\hat{\theta}})/(\theta^{\hat{k}} \cdot \Gamma(\hat{k}))$, where $\hat{k} = \mathbb{E}^2(\hat{\mathcal{T}}_{CI}|\mathcal{D})/\mathbb{V}ar(\hat{\mathcal{T}}_{CI}|\mathcal{D})$, and $\hat{\theta} = \mathbb{V}ar(\hat{\mathcal{T}}_{CI}|\mathcal{D})/\mathbb{E}(\hat{\mathcal{T}}_{CI}|\mathcal{D})$. We propose to approximate the null distribution with the mean and variance in the following theorem.

**Theorem 5** (Null Distribution Approximation). *Under the null hypothesis $\mathcal{H}_0$ ($X$ and $Y$ are conditionally independent given $Z$), we have*

$$\mathbb{E}(\hat{\mathcal{T}}_{CI}|\mathcal{D}) = \text{tr}(\mathcal{C}_{\ddot{X}|Z}) \cdot \text{tr}(\mathcal{C}_{Y|Z}),$$
$$\mathbb{V}ar(\hat{\mathcal{T}}_{CI}|\mathcal{D}) = 2\|\mathcal{C}_{\ddot{X}|Z}\|_F^2 \cdot \|\mathcal{C}_{Y|Z}\|_F^2, \tag{8}$$

*where $\mathcal{C}_{\ddot{X}|Z} = \frac{1}{n} R_{\ddot{X}|Z}^T \boldsymbol{H}\boldsymbol{H} R_{\ddot{X}|Z}$, $\mathcal{C}_{Y|Z} = \frac{1}{n} R_{Y|Z}^T \boldsymbol{H}\boldsymbol{H} R_{Y|Z}$, and $\text{tr}(\cdot)$ means the trace operator.*

For testing the conditional independence $X \perp\!\!\!\perp Y|Z$, in this paper, we only deal with the scenarios where $X$ and $Y$ each contain a single variable while $Z$ could contain a single variable, multiple variables, or be empty. When $Z$ is empty, the test becomes the federated unconditional independent test (FUIT), as a special case. We provide more details about FUIT in Appendix A5.

## 3.2 FEDERATED INDEPENDENT CHANGE PRINCIPLE (FICP)

As described in Lemma 2, we can use independent change principle (ICP) to evaluate the dependence between two changing causal models. However, existing ICP (Huang et al., 2020) heavily relies on the kernel matrix to calculate the normalized HSIC. It may be challenging for decentralized data because the off-diagonal entries of kernel matrix require the raw data from different clients, which violates the data privacy in federated learning. Motivated by that, we propose to estimate the normalized HSIC with the following theorem.

**Theorem 6** (Federated Independent Change Principle). *In order to check whether two causal models change independently across different domains, we can estimate the dependence by*

$$\hat{\triangle}_{X \to Y} = \frac{\|\mathcal{C}_{X,\tilde{Y}}^*\|_F^2}{\text{tr}(\mathcal{C}_X^*) \cdot \text{tr}(\mathcal{C}_{\tilde{Y}}^*)}, \quad \hat{\triangle}_{Y \to X} = \frac{\|\mathcal{C}_{Y,\tilde{X}}^*\|_F^2}{\text{tr}(\mathcal{C}_Y^*) \cdot \text{tr}(\mathcal{C}_{\tilde{X}}^*)}, \tag{9}$$

*where $\tilde{Y} \triangleq (Y|X)$, $\tilde{X} \triangleq (X|Y)$, $\mathcal{C}_{X,\tilde{Y}}^*$ and $\mathcal{C}_{Y,\tilde{X}}^*$ are specially-designed covariance matrices, and $\mathcal{C}_X^*, \mathcal{C}_Y^*, \mathcal{C}_{\tilde{X}}^*$ and $\mathcal{C}_{\tilde{Y}}^*$ are specially-designed variance matrices.*

## 3.3 IMPLEMENTING FCIT AND FICP WITH SUMMARY STATISTICS

More details are given about how to implement FCIT and FICP with summary statistics. The procedures at the clients and the server are shown in Algorithm 1. Each client needs to calculate its local sample size and covariance tensor, which are aggregated into summary statistics at the server.

The summary statistics contain two parts: total sample size $n$ and covariance tensor $\mathcal{C}_{\mathcal{T}}$. With the summary statistics as a proxy, we can substitute the raw data at each client for FCD. The global

---

**Algorithm 1** FedCDH: Federated Causal Discovery from Heterogeneous Data

---

**Input:** data matrix $\mathcal{D}_k \in \mathbb{R}^{n_k \times d}$ at each client, $k, \mho \in \{1, \dots, K\}$
**Output:** a causal graph $\mathcal{G}$
    **Client executes:**
1: *(Summary Statistics Calculation)* For each client $k$, use the local data $\mathcal{D}_k$ to get the sample size $n_k$ and calculate the covariance tensor $\mathcal{C}_{\mathcal{T}_k}$, and send them to the server.
    **Server executes:**
2: *(Summary Statistics Construction)* Construct the summary statistics by summing up the local sample sizes and the local covariance tensors: $n = \sum_{k=1}^{K} n_k$, $\mathcal{C}_{\mathcal{T}} = \sum_{k=1}^{K} \mathcal{C}_{\mathcal{T}_k}$.
3: *(Augmented Graph Initialization)* Build a completely undirected graph $\mathcal{G}_0$ on the extended variable set $\boldsymbol{V} \cup \{\mho\}$, where $\boldsymbol{V}$ denotes the observed variables and $\mho$ is surrogate variable.
4: *(Federated Conditional Independence Test)* Conduct the federated conditional independence test based on the summary statistics, for skeleton discovery on augmented graph and direction determination with one changing causal module. In the end, get an intermediate graph $\mathcal{G}_1$.
5: *(Federated Independent Change Principle)* Conduct the federated independent change principle based on the summary statistics, for direction determination with two changing causal modules. Ultimately, output the causal graph $\mathcal{G}$.

---

statistics are decomposable because they could be obtained by simply summing up the local ones, such as $n = \sum_{k=1}^{K} n_k$ and $\mathcal{C}_{\mathcal{T}} = \frac{1}{n} \sum_{k=1}^{K} n_k \mathcal{C}_{\mathcal{T}_k}$. Specifically, we incorporate the random Fourier features (Rahimi & Recht, 2007), because they have shown competitive performances to approximate the continuous shift-invariant kernels. According to the following Lemma, we could derive a decomposable covariance matrix from an indecomposable kernel matrix via random features.

**Lemma 7** (Estimating Covariance Matrix from Kernel Matrix). *Assuming there are $n$ i.i.d. samples for the centralized kernel matrices $\tilde{\boldsymbol{K}}_{\boldsymbol{x}}, \tilde{\boldsymbol{K}}_{\boldsymbol{y}}, \tilde{\boldsymbol{K}}_{\boldsymbol{x,y}}$ and the covariance matrix $\mathcal{C}_{\boldsymbol{x,y}}$, we have*

$$
\begin{aligned}
\mathrm{tr}(\tilde{\boldsymbol{K}}_{\boldsymbol{x,y}}) &\approx \mathrm{tr}(\tilde{\phi}_{\boldsymbol{w}}(\boldsymbol{x})\tilde{\phi}_{\boldsymbol{w}}(\boldsymbol{y})^T) = \mathrm{tr}(\tilde{\phi}_{\boldsymbol{w}}(\boldsymbol{y})^T \tilde{\phi}_{\boldsymbol{w}}(\boldsymbol{x})) = n\,\mathrm{tr}(\mathcal{C}_{\boldsymbol{x,y}}), \\
\mathrm{tr}(\tilde{\boldsymbol{K}}_{\boldsymbol{x}}\tilde{\boldsymbol{K}}_{\boldsymbol{y}}) &\approx \mathrm{tr}(\tilde{\phi}_{\boldsymbol{w}}(\boldsymbol{x})\tilde{\phi}_{\boldsymbol{w}}(\boldsymbol{x})^T \tilde{\phi}_{\boldsymbol{w}}(\boldsymbol{y})\tilde{\phi}_{\boldsymbol{w}}(\boldsymbol{y})^T) = \|\tilde{\phi}_{\boldsymbol{w}}(\boldsymbol{x})^T \tilde{\phi}_{\boldsymbol{w}}(\boldsymbol{y})\|^2 = n^2 \|\mathcal{C}_{\boldsymbol{x,y}}\|^2,
\end{aligned}
\tag{10}
$$

*where $\boldsymbol{x}, \boldsymbol{y} \in \mathbb{R}^n$, $\tilde{\boldsymbol{K}}_{\boldsymbol{x}}, \tilde{\boldsymbol{K}}_{\boldsymbol{y}}, \tilde{\boldsymbol{K}}_{\boldsymbol{x,y}} \in \mathbb{R}^{n \times n}$, $\mathcal{C}_{\boldsymbol{x,y}} \in \mathbb{R}^{h \times h}$, $\tilde{\phi}_{\boldsymbol{w}}(\boldsymbol{x}) \in \mathbb{R}^{n \times h}$ is the centralized random feature, $\tilde{\phi}_{\boldsymbol{w}}(\boldsymbol{x}) = \boldsymbol{H}\phi_{\boldsymbol{w}}(\boldsymbol{x})$, $\phi_{\boldsymbol{w}}(\boldsymbol{x}) \triangleq \sqrt{\frac{2}{h}}[\cos(w_1\boldsymbol{x}+b_1), \dots, \cos(w_h\boldsymbol{x}+b_h)]^T$ and $\phi_{\boldsymbol{w}}(\boldsymbol{x}) \in \mathbb{R}^{n \times h}$, and similarly for $\tilde{\phi}_{\boldsymbol{w}}(\boldsymbol{y})$ and $\phi_{\boldsymbol{w}}(\boldsymbol{y})$. $\boldsymbol{w}$ is drawn from $\mathbb{P}(\boldsymbol{w})$ and $\boldsymbol{b}$ is drawn uniformly from $[0, 2\pi]$.*

In this paper, we use random features to approximate the Gaussian kernel for continuous variables and the delta kernel for discrete variables such as the surrogate variable $\mho$. It is important to note that this surrogate variable $\mho$ is essentially a discrete variable (more specifically, a categorical variable, with no numerical order among different values), and a common approach to deal with such discrete variables is to use delta kernel. Notice that $\mathcal{C}_{\boldsymbol{x,y}}$ denotes the covariance matrix for variable sets $\boldsymbol{x}$ and $\boldsymbol{y}$, which is sample-wise decomposable because $\mathcal{C}_{\boldsymbol{x,y}} = \frac{1}{n} \sum_{k=1}^{K} n_k \mathcal{C}_{\boldsymbol{x}_k, \boldsymbol{y}_k}$, where $\mathcal{C}_{\boldsymbol{x}_k, \boldsymbol{y}_k}$ corresponds to the local covariance matrix of variable sets $\boldsymbol{x}_k$ and $\boldsymbol{y}_k$ at the $k$-th client. Here, we have $\boldsymbol{x}_k, \boldsymbol{y}_k \in \mathbb{R}^{n_k}$, $\mathcal{C}_{\boldsymbol{x}_k, \boldsymbol{y}_k} \in \mathbb{R}^{h \times h}$. In the augmented graph, there are $d' = d+1$ variables ($d$ observed variables and one surrogate variable), thus we could construct a global covariance tensor $\mathcal{C}_{\mathcal{T}} \in \mathbb{R}^{d' \times d' \times h \times h}$ by summing up the local ones $\mathcal{C}_{\mathcal{T}_k} \in \mathbb{R}^{d' \times d' \times h \times h}$.

**Theorem 8** (Sufficiency of Summary Statistics). *The summary statistics, consisting of total sample size $n$ and covariance tensor $\mathcal{C}_{\mathcal{T}}$, are sufficient to represent all the statistics for FCD, including $\mathcal{T}_{CI}$ in Eq. 7, $\mathbb{E}(\hat{\mathcal{T}}_{CI}|\mathcal{D})$ and $\mathbb{V}ar(\hat{\mathcal{T}}_{CI}|\mathcal{D})$ in Eq. 8, and $\hat{\triangle}_{X \to Y}$ and $\hat{\triangle}_{Y \to X}$ in Eq. 9.*

According to the above theorem, with the total sample size $n$ and the global covariance tensor $\mathcal{C}_{\mathcal{T}}$ at the server, it is sufficient to conduct FCIT and FICP in the FCD procedures. More details about skeleton discovery and direction determination rules will be given in Appendix A6.

## 3.4 COMMUNICATION COSTS AND SECURE COMPUTATIONS

We propose to construct summary statistics without directly sharing the raw data, which has already preserved the data privacy to some extent. The original sample size of raw data is in $\mathbb{R}^{n \times d'}$, where

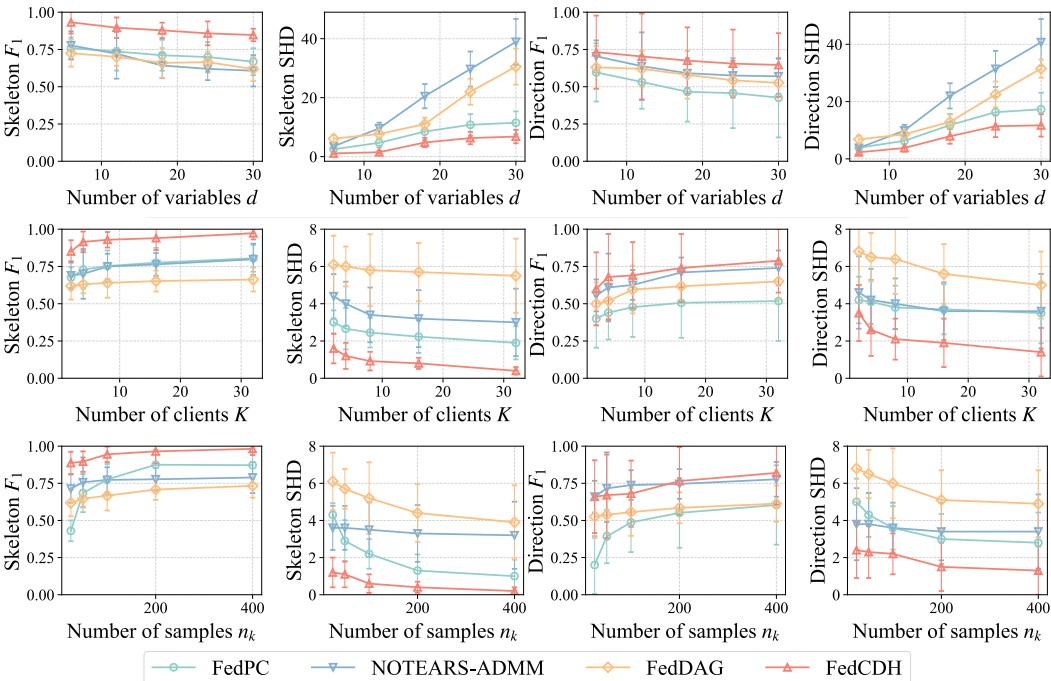

Figure 3: Results of synthetic dataset on linear Gaussian model. By rows, we evaluate varying number of variables $d$, varying number of clients $K$, and varying number of samples $n_k$. By columns, we evaluate Skeleton $F_1$ ($\uparrow$), Skeleton SHD ($\downarrow$), Direction $F_1$ ($\uparrow$) and Direction SHD ($\downarrow$).

we assume $n \gg d', h$. The constructed covariance tensor is in dimension $\mathbb{R}^{d' \times d' \times h \times h}$, which could significantly reduce the communication costs when the sample size $n$ is large enough and the hidden dimension $h$ is small enough. Furthermore, if each client is required to not directly share the local summary statistics, one can incorporate some standard secure computation techniques, such as secure multiparty computation (Cramer et al., 2015), which allows different clients to collectively compute a function over their inputs while keeping them private, or homomorphic encryption (Acar et al., 2018), which enables complex mathematical operations to process encrypted data without compromising the encryption. Please refer to Goryczka & Xiong (2015) for more about secure computation. It is worth noting that some secure computation techniques can introduce significant computation overhead. To further enhance privacy protection and computational efficiency, it would be beneficial to further improve our proposed method and we leave it for future explorations.

## 4 EXPERIMENTS

To evaluate the efficacy of our proposed method, we conduct extensive experiments on both synthetic and real-world datasets. For the synthetic datasets, we consider the linear Gaussian model and general functional model to show that our method can handle arbitrary functional causal models. We ensure that all synthetic datasets have some changing causal models, meaning that they are heterogeneous data. To show the wide applicability of our method, we run two real-world datasets, fMRI Hippocampus (Poldrack et al., 2015) and HK Stock Market datasets (Huang et al., 2020).

**Synthetic Datasets.** The true DAGs are simulated by Erdös-Rényi model (Erdős et al., 1960) with the number of edges equal to the number of variables. We randomly select 2 variables out of $d$ variables to be changing across clients. For the changing causal model, we generate according to the SCM: $V_i = \sum_{V_j \in \mathrm{PA}_i} \hat{\sigma}_{ij}^k f_i(V_j) + \hat{\gamma}^k \epsilon_i$, where $V_j \in \mathrm{PA}_i$ is the direct cause of $V_i$. The causal strength $\hat{\sigma}_{ij}^k$ and the parameter $\hat{\gamma}^k$ change across different client with index $k$, which are uniformly sampled from $\mathcal{U}(0.5, 2.5)$ and $\mathcal{U}(1, 3)$, respectively. We separately generate the data for each domain with different causal models and then combine them together. For the fixed causal model, we generate according to $V_i = \sum_{V_j \in \mathrm{PA}_i} \hat{\sigma}_{ij} f_i(V_j) + \epsilon_i$. We consider the linear Gaussian model with non-equal

noise variances and the general functional model. For linear Gaussian model, $f_i(V_j)=V_j$ and $\epsilon_i$ are sampled from Gaussian distribution with a non-equal variance which is uniformly sampled from $\mathcal{U}(1,2)$. For general functional model, $f_i$ is randomly chosen from linear, square, sinc, and tanh functions, and $\epsilon_i$ follows uniform distribution $\mathcal{U}(-0.5, 0.5)$ or Gaussian distribution $\mathcal{N}(0,1)$.

We compare our FedCDH method with other FCD baselines, such as NOTEARS-ADMM (for linear case) (Ng & Zhang, 2022), NOTEARS-MLP-ADMM (for non-linear case) (Ng & Zhang, 2022), FedDAG (Gao et al., 2022) and FedPC (Huang et al., 2022). We consider these baselines mainly because of their publicly available implementations. We evaluate both the undirected skeleton and the directed graph, denoted by "Skeleton" and "Direction" in the Figures. We use the structural Hamming distance (SHD), $F_1$ score, precision, and recall as evaluation criteria. We evaluate variable $d \in \{6, 12, 18, 24, 30\}$ while fixing other variables such as $K{=}10$ and $n_k{=}100$. We set client $K{\in}\{2, 4, 8, 16, 32\}$ while fixing others such as $d{=}6$ and $n_k{=}100$. We let the sample size in one client $n_k{\in}\{25, 50, 100, 200, 400\}$ while fixing other variables such as $d{=}6$ and $K{=}10$. Following the setting of previous works such as (Ng & Zhang, 2022), we set the sample size of each client to be equal, although our method can handle both equal and unequal sample size per client. For each setting, we run 10 instances with different random seeds and report the means and standard deviations. The results of $F_1$ score and SHD are given in Figure 3 and Figure A3 for two models, where our FedCDH method generally outperforms the other methods. Although we need large sample properties in the proof of Theorem 4, in practice we only have finite samples. According to the experiment of varying samples, we can see that with more samples the performance of our method is getting better. More analysis including the implementation details, the results of the precision and recall, the analysis of computational time, and the hyperparameter study, the statistical significance test, and the evaluation on graph density are provided in Appendix A7.

**Real-world Datasets.** We evaluate our method and the baselines on two real-world dataset, fMRI Hippocampus (Poldrack et al., 2015) and HK Stock Market datasets (Huang et al., 2020). *(i)* fMRI Hippocampus dataset contains signals from $d{=}6$ separate brain regions: perirhinal cortex (PRC), parahippocampal cortex (PHC), entorhinal cortex (ERC), subiculum (Sub), CA1, and CA3/Dentate Gyrus (DG) in the resting states on the same person in 84 successive days. The records for each day can be regarded as one domain, and there are 518 samples for each domain. We select $n_k{=}100$ samples for each day and select $K{\in}\{4, 8, 16, 32, 64\}$ days for evaluating varying number of clients. We select $K{=}10$ days and select $n_k{\in}\{25, 50, 100, 200, 400\}$ samples for evaluating varying number of samples. *(ii)* HK Stock Market dataset contains $d{=}10$ major stocks in Hong Kong stock market, which records the daily closing prices from 10/09/2006 to 08/09/2010. Here one day can be also seen as one domain. We set the number of clients to be $K{\in}\{2, 4, 6, 8, 10\}$ while randomly select $n_k{=}100$ samples for each client. All other settings are following previous one by default. More dataset information, implementation details, results and analysis are provided in Appendix A8.

## 5 DISCUSSION AND CONCLUSION

**Discussion.** *(i) Strengths:* First of all, by formulating our summary statistics, the requirement for communication between the server and clients is restricted to only one singular instance, thereby substantially reducing the communication times. This is a marked improvement over other baseline methods that necessitate iterative communications. Additionally, the utilization of a surrogate variable enhances our capability to handle heterogeneous data. Furthermore, leveraging the non-parametric characteristics of our proposed FCIT and FICP, our FedCDH method can adeptly manage arbitrary functional causal models. *(ii) Limitations:* Firstly, the efficiency of our summary statistics in reducing communication costs may not be considerable when the sample size $n$ is small or the hidden dimension $h$ is large. Secondly, our method is designed specifically for horizontally-partitioned federated data, hence it cannot be directly applied to vertically-partitioned federated data.

**Conclusion.** This paper has put forth a novel constraint-based federated causal discovery method called FedCDH, demonstrating broad applicability across arbitrary functional causal models and heterogeneous data. We construct the summary statistics as a stand-in for raw data, ensuring the protection of data privacy. We further propose FCIT and FICP for skeleton discovery and direction determination. The extensive experiments, conducted on both synthetic and real-world datasets, underscore the superior performance of our method over other baseline methods. For future research, we will enhance our method to address more complex scenarios, such as vertically-partitioned data.

ACKNOWLEDGEMENT

This material is based upon work supported by the AI Research Institutes Program funded by the National Science Foundation under AI Institute for Societal Decision Making (AI-SDM), Award No. 2229881. The project is also partially supported by the National Institutes of Health (NIH) under Contract R01HL159805, and grants from Apple Inc., KDDI Research Inc., Quris AI, and Infinite Brain Technology.

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

*Appendix for*

## "Federated Causal Discovery from Heterogeneous Data"

Appendix organization:

## A1 SUMMARY OF SYMBOLS

In order to improve the readability of our paper, we summarize the most important symbols and their meanings throughout the paper, as shown in Table A1.

Table A1: Summary of symbols

| Symbol | Meaning | | Symbol | Meaning |
|---|---|---|---|---|
| $d$ | the number of observed variables. | | $n$ | the total sample size of all clients. |
| $K$ | the number of clients. | | $\mathbf{V}$ | the data matrix, $\mathbf{V} \in \mathbb{R}^{n \times d}$. |
| $k$ | the client index, $k \in \{1, ..., K\}$. | | $n_k$ | the sample size of $k$-th client. |
| $\eth$ | domain index, $\eth \in \{1, ..., K\}$. | | $V_i$ | the $i$-th variable, $i \in \{1, ..., d\}$. |
| $f_i(\cdot)$ | the causal function of variable $V_i$. | | $\mathrm{PA}_i$ | the parents of $V_i$. |
| $\epsilon_i$ | the noise term of $V_i$. | | $\tilde{\psi}$ | a set of "pseudo confounders". |
| $\theta_i$ | the effective parameter of $V_i$. | | $L$ | the number of "pseudo confounders". |
| $\mathcal{G}, \mathcal{G}_{obs}$ | the causal graph with $d$ variables. | | $\mathcal{G}_{aug}$ | the augmented graph with $d+1$ variables. |
| $X, Y, Z$ | a set of random variables. | | $k_{\mathcal{X}}, k_{\mathcal{Y}}, k_{\mathcal{Z}}$ | the positive definite kernels. |
| $\mathcal{X}, \mathcal{Y}, \mathcal{Z}$ | the domains for the variables. | | $\mathcal{H}_{\mathcal{X}}, \mathcal{H}_{\mathcal{Y}}, \mathcal{H}_{\mathcal{Z}}$ | the reproducing kernel Hilbert spaces. |
| $\ddot{X}$ | $\ddot{X} \triangleq (X, Z)$. | | $\hat{\triangle}$ | the normalized HSIC. |
| $\Sigma$ | the cross-covariance operator in infinite dimension. | | $\mathcal{C}$ | the cross-covariance matrix in finite dimension. |
| $h$ | the number of hidden features/mapping functions. | | $f(\cdot), g(\cdot), q(\cdot)$ | the mapping functions. |
| $\mathcal{F}$ | the function spaces. | | $u, v$ | the regression coefficients. |
| $\ddot{A}$ | $\ddot{A} = f(\ddot{X}) \in \mathbb{R}^{n \times h}$. | | $B$ | $B = g(Y) \in \mathbb{R}^{n \times h}$. |
| $\gamma$ | the ridge parameter. | | $\boldsymbol{I}$ | the identity matrix. |
| $\tilde{\boldsymbol{K}}$ | the centralized kernel matrix. | | $\boldsymbol{H}$ | the matrix for centralization, $\boldsymbol{H} = \boldsymbol{I} - \frac{1}{n}\mathbf{1}\mathbf{1}^T$. |
| $\mathcal{T}_{CI}$ | the test statistic for conditional independence. | | $R$ | the residual for ridge regression. |
| $\alpha$ | the standard Gaussian variable. | | $\alpha^2$ | the $\chi_1^2$ variable. |
| $\lambda$ | the nonzero eigenvalues. | | $\tilde{\mathcal{T}}_{CI}$ | the asymptotic statistic. |
| $\hat{k}, \hat{\theta}$ | the parameters for Gamma distribution $\Gamma(\hat{k}, \hat{\theta})$. | | $\mathcal{C}^*$ | the specially-designed covariance matrix. |
| $\mathrm{tr}(\cdot)$ | the trace operator. | | $d'$ | $d' = d + 1$ (plus one surrogate variable). |
| $\mathcal{C}_{\mathcal{T}}$ | global covariance tensor, $\mathcal{C}_{\mathcal{T}} \in \mathbb{R}^{d' \times d' \times h \times h}$. | | $\mathcal{C}_{\mathcal{T}_k}$ | local covariance tensor, $\mathcal{C}_{\mathcal{T}_k} \in \mathbb{R}^{d' \times d' \times h \times h}$. |
| $\boldsymbol{w}$ | the coefficients for random features. | | $\boldsymbol{b}$ | the intercepts for random features. |

## A2 RELATED WORKS

**Causal Discovery.** In general, there are mainly three categories of methods for causal discovery (CD) from observed data (Spirtes & Zhang, 2016): constraint-based methods, score-based methods and function-based methods. Constraint-based methods utilize the conditional independence test (CIT) to learn a skeleton of the directed acyclic graph (DAG), and then orient the edges upon the skeleton. Such methods contain Peter-Clark (PC) algorithm (Spirtes & Zhang, 2016) and Fast Causal Inference (FCI) algorithm (Spirtes, 2001). Some typical CIT methods include kernel-based independent conditional test (Zhang et al., 2012) and approximate kernel-based conditional independent test (Strobl et al., 2019). Score-based methods use a score function and a greedy search method to learn a DAG with the highest score by searching all possible DAGs from the data, such as Greedy Equivalent Search (GES) (Chickering, 2002). Within the score-based category, there is a continuous optimization-base subcategory attracting increasing attention. NOTEARS (Zheng et al., 2018) firstly reformulates the DAG learning process as a continuous optimization problem and solves it using gradient-based method. NOTEARS is designed under the assumption of the linear relations between variables. Subsequent works have extended NOTEARS to handle nonlinear cases via deep neural networks, such as DAG-GNN (Yu et al., 2019) and DAG-NoCurl (Yu et al., 2021). ENCO (Lippe et al., 2022) presents an efficient DAG discovery method for directed acyclic causal graphs utilizing both observational and interventional data. AVCI (Lorch et al., 2022) infers causal structure by performing amortized variational inference over an arbitrary data-generating distribution. These continuous-optimization-based methods might suffer from various technical issues, including convergence (Wei et al., 2020; Ng et al., 2022), nonconvexity (Ng et al., 2023), and sensitivity to

data standardization (Reisach et al., 2021). Function-based methods rely on the causal asymmetry property, including the linear non-Gaussion model (LiNGAM) (Shimizu et al., 2006), the additive noise model (Hoyer et al., 2008), and the post-nonlinear causal model (Zhang & Hyvarinen, 2012).

**Causal Discovery from Heterogeneous Data.** Most of the causal discovery methods mentioned above usually assume that the data is independently and identically distributed (i.i.d.). However, in practical scenarios, distribution shift is possibly occurring across datasets, which can be changing across different domains or over time, as featured by heterogeneous or non-stationary data (Huang et al., 2020). To tackle the issue of changing causal models, one may try to find causal models on sliding windows for non-stationary data (Calhoun et al., 2014), and then compare them. Improved versions include the regime aware learning algorithm to learn a sequence of Bayesian networks that model a system with regime changes (Bendtsen, 2016). Such methods may suffer from high estimation variance due to sample scarcity, large type II errors, and a large number of statistical tests. Some methods aim to estimate the time-varying causal model by making use of certain types of smoothness of the change (Huang et al., 2015), but they do not explicitly locate the changing causal modules. Several methods aim to model time-varying time-delayed causal relations (Xing et al., 2010), which can be reduced to online parameter learning because the direction of the causal relations is given (i.e., the past influences the future). Moreover, most of these methods assume linear causal models, limiting their applicability to complex problems with nonlinear causal relations. In particular, a nonparametric constraint-based method to tackle this causal discovery problem from non-stationary or heterogeneous data, called CD-NOD (Huang et al., 2020), was recently proposed, where the surrogate variable was introduced, written as smooth functions of time or domain index. The first model-based method was proposed for heterogeneous data in the presence of cyclic causality and confounders, named CHOD (Zhou et al., 2022). Saeed et al. (Saeed et al., 2020) provided a graphical representation via the mixture DAG of distributions that arise as mixtures of causal DAGs.

**Federated Causal Discovery.** A two-step procedure was adopted (Gou et al., 2007) to learn a DAG from horizontally partitioned data, which firstly estimated the structures independently using each client's local dataset, and secondly applied further conditional independence test. Instead of using statistical test in the second step, a voting scheme was used to pick those edges identified by more than half of the clients (Na & Yang, 2010). These methods leverage only the final graphs independently estimated from each local dataset, which may lead to suboptimal performance as the information exchange may be rather limited. Furthermore, (Samet & Miri, 2009) developed a privacy-preserving method based on secure multiparty computation, but was limited to the discrete case. For vertically partitioned data, (Yang et al., 2019) constructed an approximation to the score function in the discrete case and adopted secure multiparty computation. (Chen et al., 2003) developed a four-step procedure that involves transmitting a subset of samples from each client to a central site, which may lead to privacy concern. NOTEARS-ADMM (Ng & Zhang, 2022) and Fed-DAG (Gao et al., 2022) were proposed for the federated causal discovery (FCD) based on continuous optimization methods. Fed-PC (Huang et al., 2022) was developed as a federated version of classical PC algorithm, however, it was developed for homogeneous data, which may lead to poor performance on heterogeneous data. DARLIS (Ye et al., 2022) utilizes the distributed annealing (Arshad & Silaghi, 2004) strategy to search for the optimal graph, while PERI (Mian et al., 2023) aggregates the results of the local greedy equivalent search (GES) (Chickering, 2002) and chooses the worst-case regret for each iteration. Fed-CD (Abyaneh et al., 2022) was proposed for both observational and interventional data based on continuous optimization. FEDC$^2$SL (Wang et al., 2023) extended $\chi^2$ test to the federated version, however, this method is restrictive on discrete variables and therefore not applicable for any continuous variables. Notice that most of these above-mentioned methods heavily rely on either identifiable functional causal models or homogeneous data distributions. These assumptions may be overly restrictive and difficult to be satisfied in real-world scenarios, limiting their diverse applicability.

## A3 DETAILS ABOUT THE CHARACTERIZATION

### A3.1 CHARACTERIZATION OF CONDITIONAL INDEPENDENCE

In this section, we will provide more details about the interpretation of $\Sigma_{\ddot{X}Y|Z}$ as formulated in Eq. 13, the definition of characteristic kernel as shown in Lemma 9, which is helpful to understand the

Lemma 1 in the main paper. We then provide the uncorrelatedness-based characterization of CI in Lemma 10.

First of all, for the random vector $(X, Y)$ on $\mathcal{X} \times \mathcal{Y}$, the cross-covariance operator from $\mathcal{H}_\mathcal{Y}$ to $\mathcal{H}_\mathcal{X}$ is defined by the relation

$$\langle f, \Sigma_{XY} g \rangle_{\mathcal{H}_\mathcal{X}} = \mathbb{E}_{XY}[f(X)g(Y)] - \mathbb{E}_X[f(X)]\mathbb{E}_Y[g(Y)], \tag{11}$$

for all $f \in \mathcal{H}_\mathcal{X}$ and $g \in \mathcal{H}_\mathcal{Y}$. Furthermore, we define the partial cross-covariance operator as

$$\Sigma_{XY|Z} = \Sigma_{XY} - \Sigma_{XZ}\Sigma_{ZZ}^{-1}\Sigma_{ZY}. \tag{12}$$

If $\Sigma_{ZZ}$ is not invertible, use the right inverse instead of the inverse. We can intuitively interpret the operator $\Sigma_{XY|Z}$ as the partial cross-covariance between $\{f(X), \forall f \in \mathcal{H}_\mathcal{X}\}$ and $\{g(Y), \forall g \in \mathcal{H}_\mathcal{Y}\}$ given $\{q(Z), \forall q \in \mathcal{H}_\mathcal{Z}\}$.

**Lemma 9** (Characteristic Kernel (Fukumizu et al., 2007))**.** *A kernel $\mathcal{K}_\mathcal{X}$ is characteristic, if the condition $\mathbb{E}_{X \sim \mathbb{P}_X}[f(X)] = \mathbb{E}_{X \sim \mathbb{Q}_X}[f(X)]$ ($\forall f \in \mathcal{H}_\mathcal{X}$) implies $\mathbb{P}_X = \mathbb{Q}_X$, where $\mathbb{P}_X$ and $\mathbb{Q}_X$ are two probability distributions of $X$. Gaussian kernel and Laplacian kernel are characteristic kernels.*

As shown in Lemma 1, if we use characteristic kernel and define $\ddot{X} \triangleq (X, Z)$, the characterization of CI could be related to the partial cross-covariance as $\Sigma_{\ddot{X}Y|Z} = 0 \iff X \perp\!\!\!\perp Y|Z$, where

$$\Sigma_{\ddot{X}Y|Z} = \Sigma_{\ddot{X}Y} - \Sigma_{\ddot{X}Z}\Sigma_{ZZ}^{-1}\Sigma_{ZY}. \tag{13}$$

Similarly, we can intuitively interpret the operator $\Sigma_{\ddot{X}Y|Z}$ as the partial cross-covariance between $\{f(\ddot{X}), \forall f \in \mathcal{H}_{\ddot{\mathcal{X}}}\}$ and $\{g(Y), \forall g \in \mathcal{H}_\mathcal{Y}\}$ given $\{q(Z), \forall q \in \mathcal{H}_\mathcal{Z}\}$.

Based on Lemma 1, we further consider a different characterization of CI which enforces the uncorrelatedness of functions in suitable spaces, which may be intuitively more appealing. Denote the probability distribution of $X$ as $\mathbb{P}_X$ and the joint distribution of $(X, Y)$ as $\mathbb{P}_{XY}$. Let $L_X^2$ be the space of square integrable functions of $X$ and $L_{XY}^2$ be that of $(X, Y)$. Specifically, $L_X^2 = \{f(X) | \mathbb{E}(f^2) < \infty\}$, and likewise for $L_{XY}^2$. Particularly, consider the following constrained $L^2$ spaces:

$$\begin{aligned}
\mathcal{S}_{\ddot{X}} &\triangleq \{f \in L_{\ddot{X}}^2 \mid \mathbb{E}(f|Z) = 0\}, \\
\mathcal{S}_{\ddot{Y}} &\triangleq \{g \in L_{\ddot{Y}}^2 \mid \mathbb{E}(g|Z) = 0\}, \\
\mathcal{S}_{Y|Z}' &\triangleq \{g' \mid g' = g(Y) - \mathbb{E}(g|Z), g \in L_Y^2\}.
\end{aligned} \tag{14}$$

They can be constructed from the corresponding $L^2$ spaces via nonlinear regression. From example, for any function $f \in L_{XZ}^2$, the corresponding function $f'$ is given by:

$$f'(\ddot{X}) = f(\ddot{X}) - \mathbb{E}(f|Z) = f(\ddot{X}) - \beta_f^*(Z), \tag{15}$$

where $\beta_f^*(Z) \in L_Z^2$ is the regression function of $f(\ddot{X})$ on $Z$. Then, we can then relate the different characterization of CI from Lemma 1 to the uncorrelatedness in the following lemma.

**Lemma 10** (Characterization of CI based on Partial Association (Daudin, 1980))**.** *Each of the following conditions are equivalent to $X \perp\!\!\!\perp Y|Z$*

$$\begin{aligned}
&(i.)\ \mathbb{E}(fg) = 0, \forall f \in \mathcal{S}_{\ddot{X}} \text{ and } \forall g \in \mathcal{S}_{\ddot{Y}}, \\
&(ii.)\ \mathbb{E}(fg') = 0, \forall f \in \mathcal{S}_{\ddot{X}} \text{ and } \forall g' \in \mathcal{S}_{Y|Z}', \\
&(iii.)\ \mathbb{E}(f\tilde{g}) = 0, \forall f \in \mathcal{S}_{\ddot{X}} \text{ and } \forall \tilde{g} \in L_{\ddot{Y}}^2, \\
&(iv.)\ \mathbb{E}(f\tilde{g}') = 0, \forall f \in \mathcal{S}_{\ddot{X}} \text{ and } \forall \tilde{g}' \in L_Y^2.
\end{aligned} \tag{16}$$

When $(X, Y, Z)$ are jointly Gaussian, the independence is equivalent to the uncorrelatedness, in other words, $X \perp\!\!\!\perp Y|Z$ is equivalent to the vanishing of the partial correlation coefficient $\rho_{XY|Z}$. We can regard the Lemma 10 as as a generalization of the partial correlation based characterization of CI.

For example, condition (*i*) means that any "residual" function of $(X, Z)$ given $Z$ is uncorrelated with that of $(Y, Z)$ given $Z$. Here we can observe the similarity between Lemma 1 and Lemma 10, except the only difference that Lemma 10 considers all functions in $L^2$ spaces, while Lemma 1 exploits the spaces corresponding to some characteristic kernels. If we restrict the function $f$ and $g'$ in condition (*ii*) to the spaces $\mathcal{H}_{\ddot{\mathcal{X}}}$ and $\mathcal{H}_{\mathcal{Y}}$, respectively, Lemma 10 is then reduced to Lemma 1.

Based on the two lemmas mentioned above plus the Lemma 1, we could further derive Lemma 3 in our main paper.

### A3.2 CHARACTERIZATION OF INDEPENDENT CHANGE

In Lemma 2 of the main paper, we provide the independent change principle (ICP) to evaluate the dependence between two changing causal models. Here, we give more details about the definition and the assigned value of normalized HSIC. A smaller value means being more independent.

**Definition 1** (Normalized HSIC (Fukumizu et al., 2007)). *Given variables $U$ and $V$, HSIC provides a measure for testing their statistical independence. An estimator of normalized HSIC is given as*

$$\text{HSIC}_{UV}^{\mathcal{N}} = \frac{\text{tr}(\tilde{M}_U \tilde{M}_V)}{\text{tr}(\tilde{M}_U)\,\text{tr}(\tilde{M}_V)}, \tag{17}$$

where $\tilde{M}_U$ and $\tilde{M}_V$ are the centralized Gram matrices, $\tilde{M}_U \triangleq HM_U H$, $\tilde{M}_V \triangleq HM_V H$, $H = I - \frac{1}{n}\mathbf{1}\mathbf{1}^T$, $I$ is $n \times n$ identity matrix and $\mathbf{1}$ is vector of $n$ ones. How to construct $M_U$ and $M_V$ will be explained in the corresponding cases below. To check whether two causal modules change independently across different domains, the dependence between $\mathbb{P}(X)$ and $\mathbb{P}(Y|X)$ and the dependence between $\mathbb{P}(Y)$ and $\mathbb{P}(X|Y)$ on the given data can be given by

$$\triangle_{X \to Y} = \frac{\text{tr}(\tilde{M}_X \tilde{M}_{Y|X})}{\text{tr}(\tilde{M}_X)\,\text{tr}(\tilde{M}_{Y|X})}, \quad \triangle_{Y \to X} = \frac{\text{tr}(\tilde{M}_Y \tilde{M}_{X|Y})}{\text{tr}(\tilde{M}_Y)\,\text{tr}(\tilde{M}_{X|Y})}. \tag{18}$$

According to CD-NOD (Huang et al., 2020), instead of working with conditional distribution $\mathbb{P}(X|Y)$ and $\mathbb{P}(Y|X)$, we could use the "joint distribution" $\mathbb{P}(X, Y)$, which is simpler, for estimation. Here we use $\underline{Y}$ instead of $Y$ to emphasize that in this constructed distribution $X$ and $Y$ are not symmetric. Then, the dependence values listed in Eq. 18 could be estimated by

$$\hat{\triangle}_{X \to Y} = \frac{\text{tr}(\tilde{M}_X \tilde{M}_{\underline{Y}X})}{\text{tr}(\tilde{M}_X)\,\text{tr}(\tilde{M}_{\underline{Y}X})}, \quad \hat{\triangle}_{Y \to X} = \frac{\text{tr}(\tilde{M}_Y \tilde{M}_{\underline{X}Y})}{\text{tr}(\tilde{M}_Y)\,\text{tr}(\tilde{M}_{\underline{X}Y})}, \tag{19}$$

where $\tilde{M}_X \triangleq HM_X H$, $M_X \triangleq \hat{\mu}_{X|\mho} \cdot \hat{\mu}_{X|\mho}^T$. Similarly, we define $\tilde{M}_Y$, $M_Y$ and $\hat{\mu}_{Y|\mho}$. According to (Huang et al., 2020), we have

$$\hat{\mu}_{X|\mho} \triangleq \phi(\mho)(\mathcal{C}_{\mho\mho} + \gamma I)^{-1}\mathcal{C}_{\mho X}, \tag{20}$$

where $\hat{\mu}_{X|\mho} \triangleq \phi(\mho)(\mathcal{C}_{\mho\mho} + \gamma I)^{-1}\mathcal{C}_{\mho X}$, $\hat{\mu}_{X|\mho}, \phi(\mho) \in \mathbb{R}^{n \times h}$, $\gamma$ is a small ridge parameter, $\phi$ represents the feature map, and $\mho$ is the surrogate variable indicating different domains or clients. Similarly, we define $\tilde{M}_Y$, $M_Y$ and $\hat{\mu}_{Y|\mho}$.

$$\hat{\mu}_{Y|\mho} \triangleq \phi(\mho)(\mathcal{C}_{\mho\mho} + \gamma I)^{-1}\mathcal{C}_{\mho Y}. \tag{21}$$

Moreover, $\tilde{M}_{\underline{Y}X} \triangleq HM_{\underline{Y}X}H$, $M_{\underline{Y}X} \triangleq \hat{\mu}_{\underline{Y}X|\mho} \cdot \hat{\mu}_{\underline{Y}X|\mho}^T$. Similarly, we define $\tilde{M}_{\underline{X}Y}$, $M_{\underline{X}Y}$ and $\hat{\mu}_{\underline{X}Y}$.

$$\begin{aligned} \hat{\mu}_{\underline{Y}X|\mho} &\triangleq \phi(\mho)(\mathcal{C}_{\mho\mho} + \gamma I)^{-1}\mathcal{C}_{\mho,(Y,X)} \\ \hat{\mu}_{\underline{X}Y|\mho} &\triangleq \phi(\mho)(\mathcal{C}_{\mho\mho} + \gamma I)^{-1}\mathcal{C}_{\mho,(X,Y)}, \end{aligned} \tag{22}$$

Eq. 19 as formulated above is helpful to further derive Theorem 5 in our main paper.

### A4 PROOFS

Here, we provide the proofs of the theorems and lemmas, including Lemma 3, Theorem 4, Theorem 5, Theorem 6, Lemma 7, and Theorem 8 in our main paper.

### A4.1 PROOF OF LEMMA 3

**Proof:** We define the covariance matrix in the null hypothesis as $\mathcal{C}_{\ddot{X}Y|Z} = \frac{1}{n}\sum_{i=1}^{n}[(\ddot{A}_i - \mathbb{E}(\ddot{A}|Z))^T(B_i - \mathbb{E}(B|Z))]$ which corresponds to the partial cross-covariance matrix with $n$ samples, $\mathcal{C}_{\ddot{X}Y|Z}\in\mathbb{R}^{h\times h}$, $\ddot{A}=f(\ddot{X})\in\mathbb{R}^{n\times h}$, $B=g(Y)\in\mathbb{R}^{n\times h}$, $\{f^j(\ddot{X})|_{j=1}^{h}\}\in\mathcal{F}_{\ddot{X}}$, $\{g^j(Y)|_{j=1}^{h}\}\in\mathcal{F}_Y$. Notice that $\mathcal{F}_{\ddot{X}}$ and $\mathcal{F}_Y$ are function spaces. $n$ and $h$ denote the number of total samples of all clients and the number of hidden features or mapping functions, respectively.

Notice that $\mathbb{E}(\ddot{A}|Z)$ and $\mathbb{E}(B|Z)$ could be non-linear functions of $Z$ which may be difficult to estimate. therefore, we would like to approximate them with linear functions. Let $q(Z)\in\mathbb{R}^{n\times h}$, $\{q^j(Z)|_{j=1}^{h}\}\in\mathcal{F}_Z$. We could estimate $\mathbb{E}(f^j|Z)$ with the ridge regression output $u_j^Tq(Z)$ under the mild conditions given below.

**Lemma 11.** (Sutherland & Schneider, 2015) *Consider performing ridge regression of $f^j$ on $Z$. Assume that (i) $\sum_{i=1}^{n}f_i^j = 0$, $f^j$ is defined on the domain of $\ddot{X}$; (ii) the empirical kernel matrix of $Z$, denoted by $\mathcal{K}_Z$, only has finite entries (i.e., $\|\mathcal{K}_Z\|_\infty < \infty$); (iii) the range of $Z$ is compact, $Z \subset \mathbb{R}^{d_Z}$. Then we have*

$$\mathbb{P}\left[|\hat{\mathbb{E}}(f^j|Z) - u_j^Tq(Z)| \geq \epsilon\right] \leq \frac{c_0}{\epsilon^2}e^{-h\epsilon^2 c_1}, \tag{23}$$

*where $\hat{\mathbb{E}}(f^j|Z)$ is the estimate of $\mathbb{E}(f^j|Z)$ by ridge regression, $c_0$ and $c_1$ are both constants that do not depend on the sample size $n$ or the number of hidden dimensions or mapping functions $h$.*

The exponential rate with respect to $h$ in the above lemma suggests we can approximate the output of ridge regression with a small number of hidden features. Moreover, we could similarly estimate $\mathbb{E}(g^j|Z)$ with $v_j^Tq(Z)$, because we could guarantee that $\mathbb{P}\left[|\hat{\mathbb{E}}(g^j|Z) - v_j^Tq(Z)| \geq \epsilon\right] \to 0$ for any fixed $\epsilon > 0$ at an exponential rate with respect to $h$.

Similar to the $L^2$ spaces in condition $(ii)$ of Lemma 10, we can consider the following condition to approximate conditional independence:

$$\mathbb{E}(\tilde{f}\tilde{g}) = 0, \forall \tilde{f} \in \tilde{\mathcal{F}}_{\ddot{X}|Z} \text{ and } \forall \tilde{g} \in \tilde{\mathcal{F}}_{Y|Z}, \text{ where}$$
$$\tilde{\mathcal{F}}_{\ddot{X}|Z} = \{\tilde{f} \mid \tilde{f}^j = f^j - \mathbb{E}(f^j|Z), f^j \in \mathcal{F}_{\ddot{X}}\}, \tag{24}$$
$$\tilde{\mathcal{F}}_{Y|Z} = \{\tilde{g} \mid \tilde{g}^j = g^j - \mathbb{E}(g^j|Z), g^j \in \mathcal{F}_Y\}.$$

According to Eq. 23, we could estimate $\mathbb{E}(f^j|Z)$ and $\mathbb{E}(g^j|Z)$ by $u_j^Tq(Z)$ and $v_j^Tq(Z)$, respectively. Thus, we can reformulate the function spaces as

$$\tilde{\mathcal{F}}_{\ddot{X}|Z} = \{\tilde{f} \mid \tilde{f}^j = f^j - u_j^Tq(Z), f^j \in \mathcal{F}_{\ddot{X}}\},$$
$$\tilde{\mathcal{F}}_{Y|Z} = \{\tilde{g} \mid \tilde{g}^j = g^j - v_j^Tq(Z), g^j \in \mathcal{F}_Y\}. \tag{25}$$

Proof ends.

### A4.2 PROOF OF THEOREM 4

**Proof:** Assume that there are $n$ i.i.d. samples for $X, Y, Z$. Let $\tilde{\boldsymbol{K}}_{\ddot{X}|Z}$ be the centralized kernel matrix, given by $\tilde{\boldsymbol{K}}_{\ddot{X}|Z}\triangleq\tilde{R}_{\ddot{X}|Z}\tilde{R}_{\ddot{X}|Z}^T=\boldsymbol{H}R_{\ddot{X}|Z}R_{\ddot{X}|Z}^T\boldsymbol{H}$, where $R_{\ddot{X}|Z}\triangleq\tilde{f}(\ddot{X})=f(\ddot{X})-u^Tq(Z)$ which can be seen as the residual after ridge regression. Similarly, We could define $\tilde{\boldsymbol{K}}_{Y|Z}\triangleq\tilde{R}_{Y|Z}\tilde{R}_{Y|Z}^T=\boldsymbol{H}R_{Y|Z}R_{Y|Z}^T\boldsymbol{H}$ and $R_{Y|Z}\triangleq\tilde{g}(Y)=g(Y)-v^Tq(Z)$. Accordingly, we let $\tilde{\boldsymbol{K}}_{\ddot{X}Y|Z}\triangleq\tilde{R}_{\ddot{X}|Z}\tilde{R}_{Y|Z}^T=\boldsymbol{H}R_{\ddot{X}|Z}R_{Y|Z}^T\boldsymbol{H}$. We set the test statistic as $\mathcal{T}_{CI}=n\|\mathcal{C}_{\ddot{X}Y|Z}\|_F^2$, where $\mathcal{C}_{\ddot{X}Y|Z}\triangleq\tilde{R}_{\ddot{X}|Z}^T\tilde{R}_{Y|Z}=\frac{1}{n}R_{\ddot{X}|Z}^T\boldsymbol{H}\boldsymbol{H}R_{Y|Z}$.

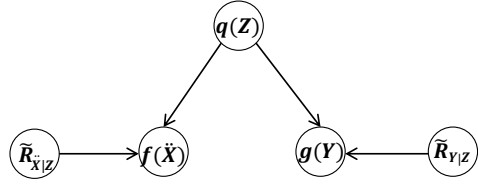

Figure A1: Given that $X \perp\!\!\!\perp Y|Z$, we could introduce the independence between $R_{\ddot{X}|Z}$ and $R_{Y|Z}$.

Let $\lambda_{\ddot{X}|Z}$ and $\lambda_{Y|Z}$ be the eigenvalues of $\tilde{\boldsymbol{K}}_{\ddot{X}|Z}$ and $\tilde{\boldsymbol{K}}_{Y|Z}$, respectively. Furthermore, we define the EVD decomposition $\tilde{\boldsymbol{K}}_{\ddot{X}|Z} = \boldsymbol{V}_{\ddot{X}|Z}\boldsymbol{\Lambda}_{\ddot{X}|Z}\boldsymbol{V}_{\ddot{X}|Z}^T$, where $\boldsymbol{\Lambda}_{\ddot{X}|Z}$ is the diagonal matrix containing non-negative eigenvalues $\lambda_{\ddot{X}|Z,i}$. Similarly, we define $\tilde{\boldsymbol{K}}_{Y|Z} = \boldsymbol{V}_{Y|Z}\boldsymbol{\Lambda}_{Y|Z}\boldsymbol{V}_{Y|Z}^T$ with eigenvalues $\lambda_{Y|Z,i}$. Let $\boldsymbol{\psi}_{\ddot{X}|Z} = [\psi_{\ddot{X}|Z,1}, \psi_{\ddot{X}|Z,2}, \ldots, \psi_{\ddot{X}|Z,n}] \triangleq \boldsymbol{V}_{Y|Z}\boldsymbol{\Lambda}_{Y|Z}^{1/2}$ and $\boldsymbol{\phi}_{Y|Z} = [\phi_{Y|Z,1}, \phi_{Y|Z,2}, \ldots, \phi_{Y|Z,n}] \triangleq \boldsymbol{V}_{Y|Z}\boldsymbol{\Lambda}_{Y|Z}^{1/2}$.

On the other hand, consider eigenvalues $\lambda_{\ddot{X}|Z,i}^*$ and eigenfunctions $u_{\ddot{X}|Z,i}$ of the kernel $k_{\ddot{X}|Z}$ w.r.t. the probablity measure with the density $\mathbb{P}(\ddot{x})$, i.e., $\lambda_{\ddot{X}|Z,i}^*$ and $u_{\ddot{X}|Z,i}$ satisfy $\int k_{\ddot{X}|Z}(\ddot{x}, \ddot{x}') \cdot u_{\ddot{X}|Z,i}(\ddot{x}) \cdot \mathbb{P}(\ddot{x})d\ddot{x} = \lambda_{\ddot{X}|Z,i}^* \cdot u_{\ddot{X}|Z,i}(\ddot{x}')$, where we assume that $u_{\ddot{X}|Z,i}$ have unit variance, i.e., $\mathbb{E}[u_{\ddot{X}|Z,i}^2(\ddot{X})] = 1$. Similarly, we define $k_{Y|Z}$, $\lambda_{Y|Z,i}^*$, and $u_{Y|Z,i}^*$. Let $\{\alpha_1, \ldots, \alpha_{n^2}\}$ denote i.i.d. standard Gaussian variables, and thus $\{\alpha_1^2, \ldots, \alpha_{n^2}^2\}$ denote i.i.d. $\chi_1^2$ variables.

**Lemma 12** (Kernel-based Conditional Independence Test (Zhang et al., 2012)). *Under the null hypothesis that $X$ and $Y$ are conditional independent given $Z$, we have that the test statistic $\mathcal{T}_{CI} \triangleq \frac{1}{n}\operatorname{tr}(\tilde{\boldsymbol{K}}_{\ddot{X}|Z}\tilde{\boldsymbol{K}}_{Y|Z})$ have the same asymptotic distribution as $\hat{\mathcal{T}}_{CI} \triangleq \frac{1}{n}\sum_{k=1}^{n^2} \tilde{\lambda}_k \cdot \alpha_k^2$, where $\tilde{\lambda}_k$ are eigenvalues of $\boldsymbol{w}\boldsymbol{w}^T$, $\boldsymbol{w} = [\boldsymbol{w}_1, \ldots, \boldsymbol{w}_n]$, with the vector $\boldsymbol{w}_t$ obtained by stacking $\boldsymbol{M}_t = [\psi_{\ddot{X}|Z,1}(\ddot{X}_t), \psi_{\ddot{X}|Z,2}(\ddot{X}_t), \ldots, \psi_{\ddot{X}|Z,n}(\ddot{X}_t)]^T \cdot [\phi_{Y|Z,1}(Y_t), \phi_{Y|Z,2}(Y_t), \ldots, \phi_{Y|Z,n}(Y_t)]$.*

In the above lemma, their test statistic is equivalent to ours, due to the fact that

$$\begin{aligned}
\frac{1}{n}\operatorname{tr}(\tilde{\boldsymbol{K}}_{\ddot{X}|Z}\tilde{\boldsymbol{K}}_{Y|Z}) &= \frac{1}{n}\operatorname{tr}(\tilde{R}_{\ddot{X}|Z}(\tilde{R}_{\ddot{X}|Z}^T\tilde{R}_{Y|Z}\tilde{R}_{Y|Z}^T)) \\
&= \frac{1}{n}\operatorname{tr}((\tilde{R}_{\ddot{X}|Z}^T\tilde{R}_{Y|Z}\tilde{R}_{Y|Z}^T)\tilde{R}_{\ddot{X}|Z}) \\
&= \frac{1}{n}\|\tilde{R}_{\ddot{X}|Z}^T\tilde{R}_{Y|Z}\|_F^2 \\
&= \frac{1}{n}\|n\mathcal{C}_{\ddot{X}Y|Z}\|_F^2 \\
&= n\|\mathcal{C}_{\ddot{X}Y|Z}\|_F^2.
\end{aligned} \tag{26}$$

However, their asymptotic distribution is different from ours. Based on their asymptotic distribution, we could go further. The first two rows of Eq. 26 hold true because of the commutative property of trace, namely, $\operatorname{tr}(AB) = BA$, refer to Lemma 6 for more details. According to the formulation of $\tilde{R}_{\ddot{X}|Z}$ and $\tilde{R}_{Y|Z}$, we have

$$\begin{cases} f(\ddot{X}) = u^T q(Z) + R_{\ddot{X}|Z} \\ g(Y) = v^T q(Z) + R_{Y|Z}. \end{cases} \tag{27}$$

Based on the above formulations, we could easily draw the causal graph as shown in Fig. A1. In particular, considering that $X$ and $Y$ are conditionally independent given $Z$, we could further determine that $R_{\ddot{X}|Z}$ and $R_{Y|Z}$ are independent, namely, we have

$$X \perp\!\!\!\perp Y|Z \iff R_{\ddot{X}|Z} \perp\!\!\!\perp R_{Y|Z}. \tag{28}$$

As $f(\ddot{X})$ and $g(Y)$ are uncorrelated, then $\mathbb{E}(\boldsymbol{w}_t) = 0$. Furthermore, the covariance is $\boldsymbol{\Sigma} = \mathbb{C}\text{ov}(\boldsymbol{w}_t) = \mathbb{E}(\boldsymbol{w}_t\boldsymbol{w}_t^T)$, where $\boldsymbol{w}$ is defined in the same way as in Lemma 12. If $R_{\ddot{X}|Z} \perp\!\!\!\perp R_{Y|Z}$, for $k \neq i$ or $l \neq j$, we denote the non-diagonal (ND) entries of $\boldsymbol{\Sigma}$ as $e_{ND}$, where

$$\begin{aligned}
e_{ND} &= \mathbb{E}[\sqrt{\lambda_{\ddot{X}|Z,i}^*\lambda_{Y|Z,j}^*\lambda_{\ddot{X}|Z,k}^*\lambda_{Y|Z,l}^*}u_{\ddot{X}|Z,i}u_{Y|Z,j}u_{\ddot{X}|Z,k}u_{Y|Z,l}] \\
&= \sqrt{\lambda_{\ddot{X}|Z,i}^*\lambda_{Y|Z,j}^*\lambda_{\ddot{X}|Z,k}^*\lambda_{Y|Z,l}^*}\mathbb{E}[u_{\ddot{X}|Z,i}u_{\ddot{X}|Z,k}]\mathbb{E}[u_{Y|Z,j}u_{Y|Z,l}] \\
&= 0.
\end{aligned} \tag{29}$$

We then denote the diagonal entries of $\boldsymbol{\Sigma}$ as $e_D$, where

$$\begin{aligned}
e_D &= \lambda_{\ddot{X}|Z,i}^*\lambda_{Y|Z,j}^*\mathbb{E}[u_{\ddot{X}|Z,i}^2]\mathbb{E}[u_{Y|Z,j}^2] \\
&= \lambda_{\ddot{X}|Z,i}^*\lambda_{Y|Z,j}^*,
\end{aligned} \tag{30}$$

which are eigenvalues of $\Sigma$. According to (Zhang et al., 2012), $\frac{1}{n}\lambda_{\ddot{X}|Z,i}$ converge in probability $\lambda^*_{\ddot{X}|Z}$. Substituting all the results into the asymptotic distribution in Lemma 12, we can get the updated asymptotic distribution

$$\hat{\mathcal{T}}_{CI} \triangleq \frac{1}{n^2} \sum_{i,j=1}^{\beta} \lambda_{\ddot{X}|Z,i} \lambda_{Y|Z,j} \alpha_{ij}^2 \quad \text{as} \quad \beta = n \to \infty. \tag{31}$$

where $\beta$ is the number of nonzero eigenvalues $\lambda_{\ddot{X}|Z}$ of the kernel matrices $\tilde{K}_{\ddot{X}|Z}$.

Consequently, $\mathcal{T}_{CI}$ and $\hat{\mathcal{T}}_{CI}$ have the same asymptotic distribution. Proof ends.

### A4.3 PROOF OF THEOREM 5

**Proof:** First of all, since $\alpha_{ij}^2$ follow the $\chi^2$ distribution with one degree of freedom, thus we have $\mathbb{E}(\alpha_{ij}^2) = 1$ and $\mathbb{V}ar(\alpha_{ij}^2) = 2$. According to the asymptotic distribution in Theorem 4 and the derivation of Lemma 7, we have

$$\begin{aligned}
\mathbb{E}(\hat{\mathcal{T}}_{CI}|\mathcal{D}) &= \frac{1}{n^2} \sum_{i,j} \lambda_{\ddot{X}|Z,i} \lambda_{Y|Z,j} \\
&= \frac{1}{n^2} \sum_i \lambda_{\ddot{X}|Z,i} \sum_j \lambda_{Y|Z,j} \\
&= \frac{1}{n^2} \operatorname{tr}(\tilde{K}_{\ddot{X}|Z}) \operatorname{tr}(\tilde{K}_{Y|Z}) \\
&= \frac{1}{n^2} \operatorname{tr}(\tilde{R}_{\ddot{X}|Z} \tilde{R}_{\ddot{X}|Z}^T) \operatorname{tr}(\tilde{R}_{Y|Z} \tilde{R}_{Y|Z}^T) \\
&= \frac{1}{n^2} \operatorname{tr}(n \cdot \mathcal{C}_{\ddot{X}|Z}) \operatorname{tr}(n \cdot \mathcal{C}_{Y|Z}) \\
&= \operatorname{tr}(\mathcal{C}_{\ddot{X}|Z}) \operatorname{tr}(\mathcal{C}_{Y|Z}),
\end{aligned} \tag{32}$$

where $\tilde{R}_{\ddot{X}|Z}$ and $\tilde{R}_{Y|Z}$ are defined in the proof of Theorem 3 above. Therefore, $\mathbb{E}(\hat{\mathcal{T}}_{CI}|\mathcal{D}) = \operatorname{tr}(\mathcal{C}_{\ddot{X}|Z}) \operatorname{tr}(\mathcal{C}_{Y|Z})$.

Furthermore, $\alpha_{ij}^2$ are independent variables across $i$ and $j$, and notice that $\operatorname{tr}(\tilde{K}_{\ddot{X}|Z}^2) = \sum_i \lambda_{\ddot{X}|Z,i}^2$, and similarly $\operatorname{tr}(\tilde{K}_{Y|Z}^2) = \sum_i \lambda_{Y|Z,i}^2$. Based on the asymptotic distribution in Theorem 4, we have

$$\begin{aligned}
\mathbb{V}ar(\hat{\mathcal{T}}_{CI}|\mathcal{D}) &= \frac{1}{n^4} \sum_{i,j} \lambda_{\ddot{X}|Z,i}^2 \lambda_{Y|Z,j}^2 \mathbb{V}ar(\alpha_{ij}^2) \\
&= \frac{2}{n^4} \sum_i \lambda_{\ddot{X}|Z,i}^2 \sum_j \lambda_{Y|Z,j}^2 \\
&= \frac{2}{n^4} \operatorname{tr}(\tilde{K}_{\ddot{X}|Z}^2) \operatorname{tr}(\tilde{K}_{Y|Z}^2).
\end{aligned} \tag{33}$$

Additionally, according to the similar rule as in Eq. 26, we have

$$\begin{aligned}
\operatorname{tr}(\tilde{K}_{\ddot{X}|Z}^2) &= \operatorname{tr}(\tilde{R}_{\ddot{X}|Z} \tilde{R}_{\ddot{X}|Z}^T \tilde{R}_{\ddot{X}|Z} \tilde{R}_{\ddot{X}|Z}^T) \\
&= \operatorname{tr}(\tilde{R}_{\ddot{X}|Z}^T \tilde{R}_{\ddot{X}|Z} \tilde{R}_{\ddot{X}|Z}^T \tilde{R}_{\ddot{X}|Z}) \\
&= \|\tilde{R}_{\ddot{X}|Z}^T \tilde{R}_{\ddot{X}|Z}\|_F^2 \\
&= \|n \cdot \mathcal{C}_{\ddot{X}|Z}\|_F^2 \\
&= n^2 \|\mathcal{C}_{\ddot{X}|Z}\|_F^2.
\end{aligned} \tag{34}$$

Similarly, we have $\operatorname{tr}(\tilde{K}_{Y|Z}^2) = n^2 \|\mathcal{C}_{Y|Z}\|_F^2$. Substituting the results into the above formulation about variance, we have $\frac{2}{n^4} \operatorname{tr}(\tilde{K}_{\ddot{X}|Z}^2) \operatorname{tr}(\tilde{K}_{Y|Z}^2) = \frac{2}{n^4} \cdot n^2 \|\mathcal{C}_{\ddot{X}|Z}\|_F^2 \cdot n^2 \|\mathcal{C}_{Y|Z}\|_F^2$. Thus, $\mathbb{V}ar(\hat{\mathcal{T}}_{CI}|\mathcal{D}) = 2 \cdot \|\mathcal{C}_{\ddot{X}|Z}\|_F^2 \cdot \|\mathcal{C}_{Y|Z}\|_F^2$. Proof ends.

### A4.4  PROOF OF THEOREM 6

**Proof:**  According to the above-mentioned formulations, we have $\tilde{M}_X \triangleq H M_X H = \tilde{\hat{\mu}}_{X|\mho} \cdot \tilde{\hat{\mu}}_{X|\mho}^T$, $\tilde{\hat{\mu}}_{X|\mho} \triangleq H \cdot \hat{\mu}_{X|\mho}$. Based on the rules of estimating covariance matrix from kernel matrix in Lemma 6, we have

$$
\begin{aligned}
\operatorname{tr}(\tilde{M}_X) &= \operatorname{tr}(\tilde{\hat{\mu}}_{X|\mho} \cdot \tilde{\hat{\mu}}_{X|\mho}^T) \\
&= \operatorname{tr}(\tilde{\hat{\mu}}_{X|\mho}^T \cdot \tilde{\hat{\mu}}_{X|\mho}) \quad\quad (35) \\
&= \operatorname{tr}((H\phi(\mho)(\mathcal{C}_{\mho\mho} + \gamma I)^{-1}\mathcal{C}_{\mho X})^T (H\phi(\mho)(\mathcal{C}_{\mho\mho} + \gamma I)^{-1}\mathcal{C}_{\mho X})) \quad\quad (36) \\
&= \operatorname{tr}(\mathcal{C}_{X\mho}(\mathcal{C}_{\mho\mho} + \gamma I)^{-1}\phi(\mho)^T H \cdot H\phi(\mho)(\mathcal{C}_{\mho\mho} + \gamma I)^{-1}\mathcal{C}_{\mho X})) \\
&= \frac{1}{n}\operatorname{tr}(\mathcal{C}_{X\mho}(\mathcal{C}_{\mho\mho} + \gamma I)^{-1}\mathcal{C}_{\mho\mho}(\mathcal{C}_{\mho\mho} + \gamma I)^{-1}\mathcal{C}_{\mho X}) \quad\quad (37) \\
&= \frac{1}{n}\operatorname{tr}(\mathcal{C}_X^*). \quad\quad (38)
\end{aligned}
$$

Eq. 35 is obtained due to the trace property of the product of the matrices, as shown in Lemma 6. Eq. 36 is substituting from Eq. 20. Here we use Eq. 38 for simple notation. We can see that it can be represented with some combinations of different covariance matrices. Similarly, we have

$$
\operatorname{tr}(\tilde{M}_Y) = \frac{1}{n}\operatorname{tr}(\mathcal{C}_{Y\mho}(\mathcal{C}_{\mho\mho} + \gamma I)^{-1}\mathcal{C}_{\mho\mho}(\mathcal{C}_{\mho\mho} + \gamma I)^{-1}\mathcal{C}_{\mho Y}) = \frac{1}{n}\operatorname{tr}(\mathcal{C}_Y^*). \quad\quad (39)
$$

Regarding the centralized Gram matrices for joint distribution, similarly we have

$$
\begin{aligned}
\operatorname{tr}(\tilde{M}_{\underline{Y}X}) &= \frac{1}{n}\operatorname{tr}(\mathcal{C}_{(Y,X),\mho}(\mathcal{C}_{\mho\mho} + \gamma I)^{-1}\mathcal{C}_{\mho\mho}(\mathcal{C}_{\mho\mho} + \gamma I)^{-1}\mathcal{C}_{\mho,(Y,X)}) = \frac{1}{n}\operatorname{tr}(\mathcal{C}_{\tilde{Y}}^*), \\
\operatorname{tr}(\tilde{M}_{\underline{X}Y}) &= \frac{1}{n}\operatorname{tr}(\mathcal{C}_{(X,Y),\mho}(\mathcal{C}_{\mho\mho} + \gamma I)^{-1}\mathcal{C}_{\mho\mho}(\mathcal{C}_{\mho\mho} + \gamma I)^{-1}\mathcal{C}_{\mho,(X,Y)}) = \frac{1}{n}\operatorname{tr}(\mathcal{C}_{\tilde{X}}^*),
\end{aligned} \quad\quad (40)
$$

where $\operatorname{tr}(\tilde{M}_{\underline{Y}X}) = \operatorname{tr}(\tilde{M}_{\underline{X}Y})$. Furthermore, based on Lemma 6 and Eq. 22, we have

$$
\begin{aligned}
\operatorname{tr}(\tilde{M}_X \tilde{M}_{\underline{Y}X}) &= \operatorname{tr}(\tilde{\hat{\mu}}_{X|\mho}\tilde{\hat{\mu}}_{X|\mho}^T \cdot \tilde{\hat{\mu}}_{\underline{Y}X|\mho}\tilde{\hat{\mu}}_{\underline{Y}X|\mho}^T) \\
&= \operatorname{tr}(\tilde{\hat{\mu}}_{X|\mho}^T\tilde{\hat{\mu}}_{\underline{Y}X|\mho}\tilde{\hat{\mu}}_{\underline{Y}X|\mho}^T \cdot \tilde{\hat{\mu}}_{X|\mho}) \quad\quad (41) \\
&= \|\tilde{\hat{\mu}}_{X|\mho}^T\tilde{\hat{\mu}}_{\underline{Y}X|\mho}\|_F^2 \\
&= \|(H\phi(\mho)(\mathcal{C}_{\mho\mho} + \gamma I)^{-1}\mathcal{C}_{\mho X})^T (H\phi(\mho)(\mathcal{C}_{\mho\mho} + \gamma I)^{-1}\mathcal{C}_{\mho,(Y,X)})\|_F^2 \quad\quad (42) \\
&= \|\mathcal{C}_{X\mho}(\mathcal{C}_{\mho\mho} + \gamma I)^{-1}\phi(\mho)^T H \cdot H\phi(\mho)(\mathcal{C}_{\mho\mho} + \gamma I)^{-1}\mathcal{C}_{\mho,(Y,X)}\|_F^2 \\
&= \|\frac{1}{n}\mathcal{C}_{X\mho}(\mathcal{C}_{\mho\mho} + \gamma I)^{-1}\mathcal{C}_{\mho\mho}(\mathcal{C}_{\mho\mho} + \gamma I)^{-1}\mathcal{C}_{\mho,(Y,X)}\|_F^2 \quad\quad (43) \\
&= \frac{1}{n^2}\|\mathcal{C}_{X,\tilde{Y}}^*\|_F^2. \quad\quad (44)
\end{aligned}
$$

Eq. 41 is obtained due to the trace property of the product of the matrices, as shown in Lemma 6. Eq. 41 is substituting from Eq. 20 and Eq. 22. Here we use Eq. 44 for simple notation. We can see that it can be represented with some combinations of different covariance matrices. Similarly, we have

$$
\begin{aligned}
\operatorname{tr}(\tilde{M}_Y \tilde{M}_{\underline{X}Y}) &= \|\frac{1}{n}\mathcal{C}_{Y\mho}(\mathcal{C}_{\mho\mho} + \gamma I)^{-1}\mathcal{C}_{\mho\mho}(\mathcal{C}_{\mho\mho} + \gamma I)^{-1}\mathcal{C}_{\mho,(X,Y)}\|_F^2 \\
&= \frac{1}{n^2}\|\mathcal{C}_{Y,\tilde{X}}^*\|_F^2.
\end{aligned} \quad\quad (45)
$$

Substituting the equations above into Eq. 19, we have

$$\hat{\triangle}_{X \to Y} = \frac{\|\mathcal{C}_{X,\tilde{Y}}^*\|_F^2}{\text{tr}(\mathcal{C}_X^*) \cdot \text{tr}(\mathcal{C}_{\tilde{Y}}^*)}, \quad \hat{\triangle}_{Y \to X} = \frac{\|\mathcal{C}_{Y,\tilde{X}}^*\|_F^2}{\text{tr}(\mathcal{C}_Y^*) \cdot \text{tr}(\mathcal{C}_{\tilde{X}}^*)}. \tag{46}$$

Proof ends.

### A4.5 PROOF OF LEMMA 7

**Proof:** First of all, we incorporate random Fourier features to approximate the kernels, because they have shown competitive performances to approximate the continuous shift-invariant kernels.

**Lemma 13** (Random Features (Rahimi & Recht, 2007)). *For a continuous shift-invariant kernel $\mathcal{K}(x,y)$ on $\mathbb{R}$, we have:*

$$\mathcal{K}(x,y) = \int_{\mathbb{R}} p(w) e^{jw(x-y)} dw = \mathbb{E}_w[\zeta_w(x)\zeta_w(y)], \tag{47}$$

*where $\zeta_w(x)\zeta_w(y)$ is an unbiased estimate of $\mathcal{K}(x,y)$ when $w$ is drawn from $p(w)$.*

Since both the probability distribution $p(w)$ and the kernel entry $\mathcal{K}(x,y)$ are real, the integral in Eq. 47 converges when the complex exponentials are replaced with cosines. Therefore, we may get a real-values mapping by:

$$\mathcal{K}(x,y) \approx \phi_w(x)^T \phi_w(y),$$

$$\phi_w(x) \triangleq \sqrt{\frac{2}{h}}[\cos(w_1 x + b_1), ..., \cos(w_h x + b_h)]^T,$$

$$\phi_w(y) \triangleq \sqrt{\frac{2}{h}}[\cos(w_1 y + b_1), ..., \cos(w_h y + b_h)]^T, \tag{48}$$

where $w$ is drawn from $p(w)$ and $b$ is drawn uniformly from $[0, 2\pi]$. $x, y, w, b \in \mathbb{R}$, and the randomized feature map $\phi_w : \mathbb{R} \to \mathbb{R}^h$. The precise form of $p(w)$ relies on the type of the shift-invariant kernel we would like to approximate. Here in this paper, we choose to approximate Gaussian kernel as one of the characteristic kernels, and thus set the probability distribution $p(w)$ to the Gaussian one. Based on Eq. 48, we have

$$\text{tr}(\tilde{\boldsymbol{K}}_{\boldsymbol{x},\boldsymbol{y}}) \approx \text{tr}(\tilde{\phi}_w(\boldsymbol{x})\tilde{\phi}_w(\boldsymbol{y})^T), \tag{49}$$

where $\boldsymbol{x}, \boldsymbol{y} \in \mathbb{R}^n$, $\tilde{\boldsymbol{K}}_{\boldsymbol{x},\boldsymbol{y}} \in \mathbb{R}^{n \times n}$, $\tilde{\phi}_w(\boldsymbol{x}) \in \mathbb{R}^{n \times h}$ is the centralized random feature, $\tilde{\phi}_w(\boldsymbol{x}) = \boldsymbol{H}\phi_w(\boldsymbol{x})$. Furthermore, benefiting from the commutative property of the trace of the product of two matrices, we have

$$\text{tr}(\tilde{\phi}_w(\boldsymbol{x})\tilde{\phi}_w(\boldsymbol{y})^T) = \text{tr}(\tilde{\phi}_w(\boldsymbol{y})^T \tilde{\phi}_w(\boldsymbol{x})), \tag{50}$$

Since each random feature is centralized, meaning the zero mean for each feature, therefore, we have:

$$\text{tr}(\tilde{\phi}_w(\boldsymbol{y})^T \tilde{\phi}_w(\boldsymbol{x})) = \text{tr}(\frac{1}{n}\mathcal{C}_{\boldsymbol{x},\boldsymbol{y}}) = \frac{1}{n}\text{tr}(\mathcal{C}_{\boldsymbol{x},\boldsymbol{y}}), \tag{51}$$

where $\mathcal{C}_{\boldsymbol{x},\boldsymbol{y}}$ is the covariance matrix for variable $x$ and $y$, $\mathcal{C}_{\boldsymbol{x},\boldsymbol{y}} \in \mathbb{R}^{h \times h}$, $h$ is the number of hidden features.

For the second formulation, we have

$$\begin{aligned}
\text{tr}(\tilde{\boldsymbol{K}}_{\boldsymbol{x}}\tilde{\boldsymbol{K}}_{\boldsymbol{y}}) &= \text{tr}[\tilde{\phi}_w(\boldsymbol{x})\tilde{\phi}_w(\boldsymbol{x})^T \tilde{\phi}_w(\boldsymbol{y})\tilde{\phi}_w(\boldsymbol{y})^T] \\
&= \text{tr}[\tilde{\phi}_w(\boldsymbol{x})(\tilde{\phi}_w(\boldsymbol{x})^T \tilde{\phi}_w(\boldsymbol{y})\tilde{\phi}_w(\boldsymbol{y})^T)] \\
&= \text{tr}[(\tilde{\phi}_w(\boldsymbol{x})^T \tilde{\phi}_w(\boldsymbol{y})\tilde{\phi}_w(\boldsymbol{y})^T)\tilde{\phi}_w(\boldsymbol{x})] \\
&= \text{tr}[\tilde{\phi}_w(\boldsymbol{x})^T \tilde{\phi}_w(\boldsymbol{y})\tilde{\phi}_w(\boldsymbol{y})^T \tilde{\phi}_w(\boldsymbol{x})] \\
&= \|\tilde{\phi}_w(\boldsymbol{x})^T \tilde{\phi}_w(\boldsymbol{y})\|_F^2 \\
&= \|n\mathcal{C}_{\boldsymbol{x},\boldsymbol{y}}\|_F^2 \\
&= n^2\|\mathcal{C}_{\boldsymbol{x},\boldsymbol{y}}\|_F^2.
\end{aligned} \tag{52}$$

Together with Eq. 49, Eq. 50, Eq. 51 and Eq. 52 formulated above, we could prove the Lemma 7 in the main paper. Proof ends.

### A4.6 Proof of Theorem 8

**Proof.** The summary statistics contain two parts: total sample size $n$ and covariance tensor $\mathcal{C}_\mathcal{T} \in \mathbb{R}^{d' \times d' \times h \times h}$. Let $\mathcal{C}_\mathcal{T}^{ij} \in \mathbb{R}^{h \times h}$ be the $(i,j)$-th entry of the covariance tensor, which denotes the covariance matrix of the $i$-th and the $j$-th variable.

With the summary statistics as a proxy, we can substitute the raw data at each client. During the procedures of causal discovery, the needed statistics include $\mathcal{T}_{CI}$ in Theorem 4, $\mathbb{E}(\hat{\mathcal{T}}_{CI}|\mathcal{D})$ and $\mathbb{V}ar(\hat{\mathcal{T}}_{CI}|\mathcal{D})$ in Theorem 5, and $\hat{\triangle}_{X \to Y}$ and $\hat{\triangle}_{Y \to X}$ in Theorem 6.

**1)** Based on the Eq. (7) in the main paper, we have

$$
\begin{aligned}
\mathcal{C}_{\ddot{X}Y|Z} &= \mathcal{C}_{\ddot{X}Y} - \mathcal{C}_{\ddot{X}Z}(\mathcal{C}_{ZZ} + \gamma I)^{-1}\mathcal{C}_{ZY} \\
&= \mathcal{C}_{(X,Z),Y} - \mathcal{C}_{(X,Z),Z}(\mathcal{C}_{ZZ} + \gamma I)^{-1}\mathcal{C}_{ZY} \\
&= (\mathcal{C}_{XY} + \mathcal{C}_{ZY}) - (\mathcal{C}_{XZ} + \mathcal{C}_{ZZ})(\mathcal{C}_{ZZ} + \gamma I)^{-1}\mathcal{C}_{ZY}
\end{aligned}
\tag{53}
$$

In this paper, we consider the scenarios where $X$ and $Y$ are single variables, and $Z$ may be a single variable, a set of variables, or empty. Assuming that $Z$ contains $L$ variables. We have

$$
\mathcal{C}_{ZY} = \sum_{i=1}^{L} \mathcal{C}_{Z_iY}, \ \ \mathcal{C}_{XZ} = \sum_{i=1}^{L} \mathcal{C}_{XZ_i}, \ \ \mathcal{C}_{ZZ} = \sum_{i=1}^{L}\sum_{j=1}^{L} \mathcal{C}_{Z_iZ_j},
\tag{54}
$$

where $\mathcal{C}_{XY}, \mathcal{C}_{Z_iY}, \mathcal{C}_{XZ_i}$, and $\mathcal{C}_{Z_iZ_j}$ are the entries of the covariance tensor $\mathcal{C}_\mathcal{T}$. According to Theorem 3, $\mathcal{T}_{CI} \triangleq n\|\mathcal{C}_{\ddot{X}Y|Z}\|_F^2$. Therefore, the summary statistics are sufficient to represent $\mathcal{T}_{CI}$.

**2)** Similar to Eq. 53, we have

$$
\mathcal{C}_{\ddot{X}|Z} = (\mathcal{C}_{XX} + 2\mathcal{C}_{XZ} + \mathcal{C}_{ZZ})(\mathcal{C}_{XZ} + \mathcal{C}_{ZZ})(\mathcal{C}_{ZZ} + \gamma I)^{-1}(\mathcal{C}_{XZ} + \mathcal{C}_{ZZ})
\tag{55}
$$

$$
\mathcal{C}_{Y|Z} = \mathcal{C}_{YY} - \mathcal{C}_{YZ}(\mathcal{C}_{ZZ} + \gamma I)^{-1}\mathcal{C}_{ZY}.
\tag{56}
$$

Substituting Eq. 54 into Eq. 55 and Eq. 56, we can also conclude that the covariance tensor is sufficient to represent $\mathcal{C}_{\ddot{X}|Z}$ and $\mathcal{C}_{Y|Z}$. In other words, the summary statistics are sufficient to represent $\mathbb{E}(\hat{\mathcal{T}}_{CI}|\mathcal{D})$ and $\mathbb{V}ar(\hat{\mathcal{T}}_{CI}|\mathcal{D})$.

**3)** As shown in section A4.3, we have

$$
\hat{\triangle}_{X \to Y} = \frac{\|\mathcal{C}_{X,\tilde{Y}}^*\|_F^2}{\mathrm{tr}(\mathcal{C}_X^*) \cdot \mathrm{tr}(\mathcal{C}_{\tilde{Y}}^*)}, \quad \hat{\triangle}_{Y \to X} = \frac{\|\mathcal{C}_{Y,\tilde{X}}^*\|_F^2}{\mathrm{tr}(\mathcal{C}_Y^*) \cdot \mathrm{tr}(\mathcal{C}_{\tilde{X}}^*)},
\tag{57}
$$

where each components can be represented as some combinations of covariance matrices, as shown in Eq. 37, Eq. 39, Eq. 40, Eq. 43, and Eq. 45. Therefore, the summary statistics are sufficient to represent $\hat{\triangle}_{X \to Y}$ and $\hat{\triangle}_{Y \to X}$.

**4)** To sum up, we could conclude that: The summary statistics, consisting of total sample size $n$ and covariance tensor $\mathcal{C}_\mathcal{T}$, are sufficient to represent all the statistics needed for federated causal discovery.

Proof ends.

## A5 Details about Federated Unconditional Independence Test

Here, we provide more details about the federated unconditional independence test (FUIT), where the conditioning set $Z$ is empty. Generally, this method follows similar theorems for federated conditional independent test (FCIT).

### A5.1   NULL HYPOTHESIS

Consider the null and alternative hypothesis

$$\mathcal{H}_0 : X \perp\!\!\!\perp Y, \quad \mathcal{H}_1 : X \not\perp\!\!\!\perp Y. \tag{58}$$

Similar to FCIT, we consider the squared Frobenius norm of the empirical covariance matrix as an approximation, given as

$$\mathcal{H}_0 : \|\mathcal{C}_{\ddot{X}Y}\|_F^2 = 0, \quad \mathcal{H}_1 : \|\mathcal{C}_{\ddot{X}Y}\|_F^2 > 0. \tag{59}$$

In this unconditional case, we set the test statistics as $\mathcal{T}_{UI} \triangleq n\|\mathcal{C}_{\ddot{X}Y}\|_F^2$, and give the following theorem.

**Theorem 14** (Federated Unconditional Independent Test). *Under the null hypothesis $\mathcal{H}_0$ ($X$ and $Y$ are independent), the test statistic*

$$\mathcal{T}_{UI} \triangleq n\|\mathcal{C}_{XY}\|_F^2, \tag{60}$$

*has the asymptotic distribution*

$$\hat{\mathcal{T}}_{UI} \triangleq \frac{1}{n^2} \sum_{i,j=1}^{L} \lambda_{X,i}\lambda_{Y,j}\alpha_{ij}^2,$$

where $\lambda_X$ and $\lambda_Y$ are the eigenvalues of $\tilde{\boldsymbol{K}}_X$ and $\tilde{\boldsymbol{K}}_Y$, respectively. Here, the proof is similar to the proof of Theorem 3, thus we refer the readers to section A4.2 for more details.

### A5.2   NULL DISTRIBUTION APPROXIMATION

We also approximate the null distribution with a two-parameter Gamma distribution, which is related to the mean and variance. Under the hypothesis $\mathcal{H}_0$ and given the sample $\mathcal{D}$, the distribution of $\hat{\mathcal{T}}_{CI}$ can be approximated by the $\Gamma(\kappa, \theta)$ distribution. Here we provide the theorem for null distribution approximation.

**Theorem 15** (Null Distribution Approximation). *Under the null hypothesis $\mathcal{H}_0$ ($X$ and $Y$ are independent), we have*

$$\begin{aligned}
\mathbb{E}(\hat{\mathcal{T}}_{UI}|\mathcal{D}) &= \mathrm{tr}(\mathcal{C}_X) \cdot \mathrm{tr}(\mathcal{C}_Y), \\
\mathbb{V}ar(\hat{\mathcal{T}}_{UI}|\mathcal{D}) &= 2\|\mathcal{C}_X\|_F^2 \cdot \|\mathcal{C}_Y\|_F^2,
\end{aligned} \tag{61}$$

Here, the proof is similar to the proof of Theorem 4, thus we refer the readers to section A4.3 for more details.

## A6   DETAILS ABOUT SKELETON DISCOVERY AND DIRECTION DETERMINATION

In this section, we will introduce how we do the skeleton discovery and direction determination during the process of federated causal discovery. All those steps are conducted on the server side. Our steps are similar to the previous method, such as CD-NOD (Huang et al., 2020), the core difference are that we develop and utilize our proposed federated conditional independent test (FCIT) and federated independent change principle (FICP).

### A6.1   SKELETON DISCOVERY.

We first conduct skeleton discovery on the augmented graph. The extra surrogate variable is introduced in order to deal with the data heterogeneity across different clients.

**Lemma 16.** *Given the Assumptions 1, 2 and 3 in the main paper, for each $V_i \in \boldsymbol{V}$, $V_i$ and $\mho$ are not adjacent in the graph if and only if they are independent conditional on some subset of $\{V_j | j \neq i\}$.*

**Proof.** If $V_i$'s causal module is invariant, which means that $\mathbb{P}(V_i|PA_i)$ remains the same for every value of $\mho$, then $V_i \perp\!\!\!\perp \mho|PA_i$. Thus, if $V_i$ and $\mho$ are not independent conditional on any subset of other variables, $V_i$'s module changes with $\mho$, which is represented by an edge between $V_i$ and $\mho$. Conversely, we assume that if $V_i$'s module changes, which entails that $V_i$ and $\mho$ are not independent given $PA_i$, then $V_i$ and $\mho$ are not independent given any other subset of $\boldsymbol{V}\backslash\{V_i\}$. Proof ends.

**Lemma 17.** *Given the Assumptions 1, 2 and 3 in the main paper, for every $V_i, V_j \in \boldsymbol{V}$, $V_i$ and $V_j$ are not adjacent if and only if they are independent conditional on some subset of $\{V_l|l \neq i, l \neq j\} \cup \{\mho\}$.*

**Proof.** The "if" direction shown based on the faithfulness assumption on $\mathcal{G}_{aug}$ and the fact that $\{\psi_l(\mho)\}_{l=1}^L \cup \{\theta_i(\mho)\}_{i=1}^d$ is a deterministic function of $\mho$. The "only if" direction is proven by making use of the weak union property of conditional independence repeatedly, the fact that all $\{\psi_l(\mho)\}_{l=1}^L$ and $\{\theta_i(\mho)\}_{i=1}^d$ are deterministic function of $\mho$, the above three assumptions, and the properties of mutual information. Please refer to (Zhang et al., 2015) for more complete proof.

With the given three assumptions in the main paper, we can do skeleton discovery.

  i) *Augmented graph initialization.* First of all, build a completely undirected graph on the extended variable set $\boldsymbol{V} \cup \{\mho\}$, where $\boldsymbol{V}$ denotes the observed variables and $\mho$ is surrogate variable.

  ii) *Changing module detection.* For each edge $\mho - V_i$, conduct the federated conditional independence test or federated unconditional independent test. If they are conditionally independent or independent, remove the edge between them. Otherwise, keep the edge and orient $\mho \rightarrow V_i$.

  iii) *Skeleton discovery.* Moreover, for each edge $V_i - V_j$, also conduct the federated independence test or federated unconditional independent test. If they are conditionally independent or independent, remove the edge between them.

In the procedures, how observed variables depend on surrogate variable $\mho$ is unknown and usually nonlinear, thus it is crucial to use a general and non-parametric conditional independent test method, which should also satisfy the federated learning constraints. Here, we utilize our proposed FCIT.

A6.2    DIRECTION DETERMINATION.

After obtaining the skeleton, we can go on with the causal direction determination. By introducing the surrogate variable $\mho$, it does not only allow us to infer the skeleton, but also facilitate the direction determinations. For each variable $V_i$ whose causal module is changing (i.e., $\mho - V_i$), in some ways we might determine the directions of every edge incident to $V_i$. Assume another variable $V_j$ which is adjacent to $V_i$, then we can determine the directions via the following rules.

  i) *Direction determination with one changing module.* When $V_j$'s causal module is not changing, we can see $\mho - V_i - V_j$ forms an unshielded triple. For practice purposes, we can take the direction between $\mho$ and $V_i$ as $\mho \rightarrow V_i$, since we let $\mho$ be the surrogate variable to indicate whether this causal module is changing or not. Then we can use the standard orientation rules (Spirtes et al., 2000) for unshielded triples to orient the edge between $V_i$ and $V_j$. (1) If $\mho$ and $V_i$ are independent conditional on some subset of $\{V_l|l \neq j\}$ which is excluding $V_j$, then the triple forms a V-structure, thus we have $\mho \rightarrow V_i \leftarrow V_j$. (2) If $\mho$ and $V_i$ are independent conditional on some subset of $\{V_l|l \neq i\} \cup \{V_j\}$ which is including $V_j$, then we have $\mho \rightarrow V_i \rightarrow V_j$. In the procedure, we apply our proposed FCIT.

  ii) *Direction determination with two changing modules.* When $V_j$'s causal module is changing, we can see there is a special confounder $\mho$ between $V_i - V_j$. First of all, as mentioned above, we can still orient $\mho \rightarrow V_i$ and $\mho \rightarrow V_j$. Then, inspired by that $P(\text{cause})$ and $P(\text{effect}|\text{cause})$ change independently, we can identify the direction between $V_i$ and $V_j$ according to Lemma 1, and we apply our proposed FICP.

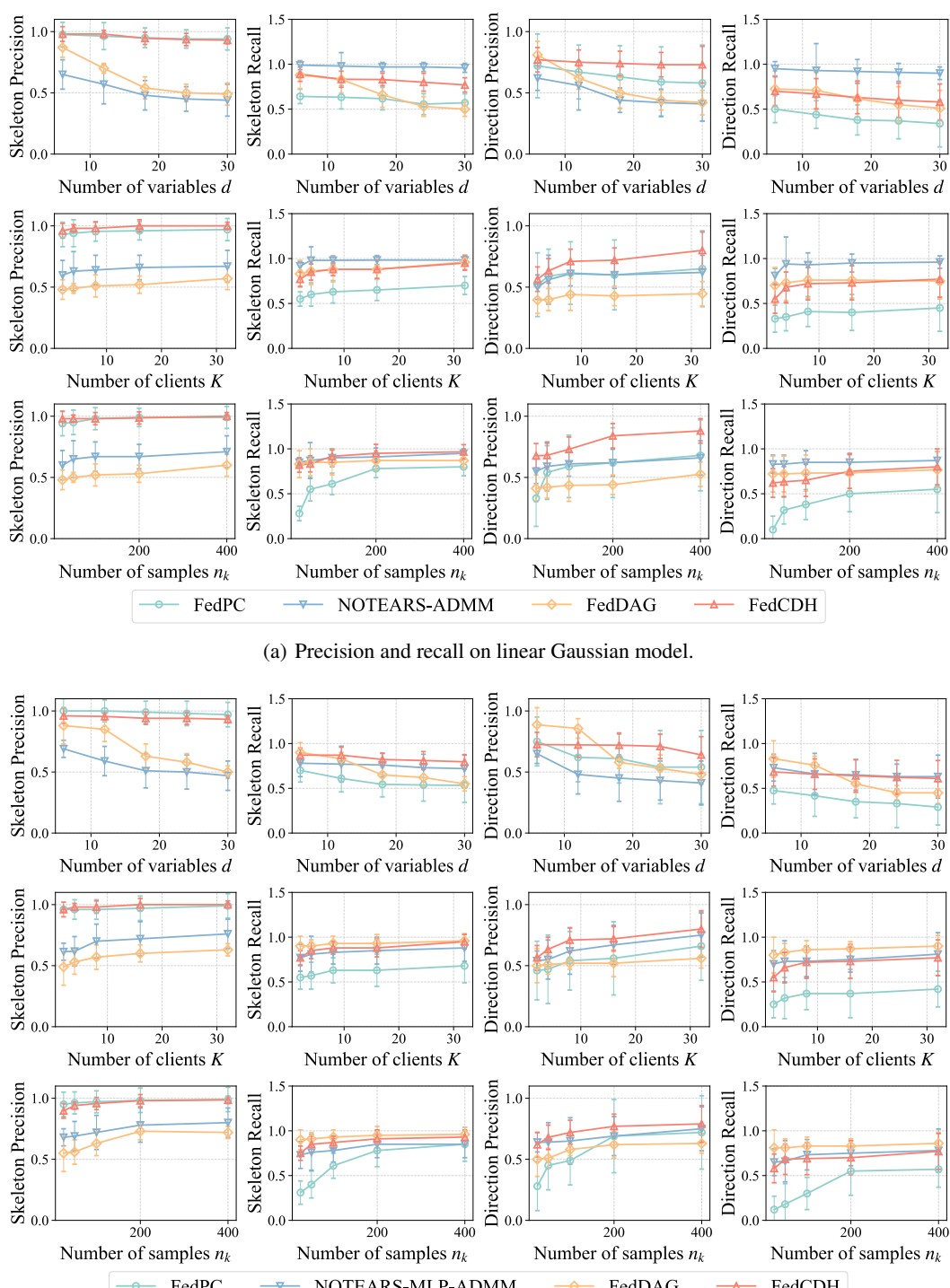

(a) Precision and recall on linear Gaussian model.

(b) Precision and recall on general functional model.

Figure A2: Results of the synthetic dataset on (a) linear Gaussian model and (b) general functional model. By rows in each subfigure, we evaluate varying number of variables $d$, varying number of clients $K$, and varying number of samples $n_k$. By columns in each subfigure, we evaluate Skeleton Precision ($\uparrow$), Skeleton Recall ($\uparrow$), Direction Precision ($\uparrow$) and Direction Recall ($\uparrow$).

## A7 DETAILS ABOUT THE EXPERIMENTS ON SYNTHETIC DATASETS

More details about the synthetic datasets are explained in this section, including the implementation details in section A7.1, the results analysis of $F_1$ and SHD in section A7.2, the complete results of precision and recall in section A7.3, the computational time analysis in section A7.4, the hyperparameter study on the number of hidden features $h$ in section A7.5, the statistical significance test for the results in section A7.6, and the evaluation on dense graph in section A7.7.

### A7.1 IMPLEMENTATION DETAILS

We provide the implementation details of our method and other baseline methods.

- FedDAG (Gao et al., 2022): Codes are available at the author's Github repository `https://github.com/ErdunGAO/FedDAG`. The hyperparameters are set by default.

- NOTEARS-ADMM and NOTEARS-MLP-ADMM (Ng & Zhang, 2022): Codes are available at the author's Github repository `https://github.com/ignavierng/notears-admm`. The hyperparameters are set by default, e.g., we set the threshold level to 0.1 for post-processing.

- FedPC (Huang et al., 2022): Although there is no public implementation provided by the author, considering that it is the only constraint-based method among all the existing works for federated causal discovery, we still compared with it. We reproduced it based on the Causal-learn package `https://github.com/py-why/causal-learn`. Importantly, we follow the paper, set the voting rate as 30% and set the significance level to 0.05.

- FedCDH (Ours): Our method is developed based on the CD-NOD (Huang et al., 2020) and KCI (Zhang et al., 2012) which are publicly available in the Causal-learn package `https://github.com/py-why/causal-learn`. We set the hyperparameter $h$ to 5, and set the significance level for FCIT to 0.05. Our source code has been appended in the Supplementary Materials.

For NOTEARS-ADMM, NOTEARS-MLP-ADMM, and FedDAG, the output is a directed acyclic graph (DAG), while FedPC and our FedCDH may output a completed partially directed acyclic graph (CPDAG). To ease comparisons, we use the simple orientation rules (Dor & Tarsi, 1992) implemented by Causal-DAG (Chandler Squires, 2018) to convert a CPDAG into a DAG. We evaluate both the undirected skeleton and the directed graph, denoted by "Skeleton" and "Direction" as shown in the Figures.

### A7.2 ANALYSIS OF $F_1$ AND SHD

We have provided the results of $F_1$ and SHD in the main paper as shown in Figure 3 and Figure A3, here we provide further discussions and analysis.

The results of linear Gaussian model are given in Figure 3 and those of general functional model are provided in Figure A3. According to the results, we observe that our FedCDH method generally outperforms all other baselines across different criteria and settings. According to the results of our method on both of the two models, when $d$ increases, the $F_1$ score decreases and the SHD increases for skeletons and directions, indicating that FCD with more variables might be more challenging. On the contrary, when $K$ and $n_k$ increase, the $F_1$ score grows and the SHD reduces, suggesting that more joint clients or samples could contribute to better performances for FCD.

In linear Gaussian model, NOTEARS-ADMM and FedPC generally outperform FedDAG. The reason may be that the front two methods were proposed for linear model while the latter one was specially proposed for nonlinear model. In general functional model, FedPC obtained the worst performance compared to other methods in direction $F_1$ score, possibly due to its strong assumptions on linear model and homogeneous data. FedDAG and NOTEARS-MLP-ADMM revealed poor results regarding SHD, the reasons may be two-fold: they assume nonlinear identifiable model, which may not well handle the general functional model; and both of them are continuous-optimization-based methods, which might suffer from various issues such as convergence and nonconvexity.

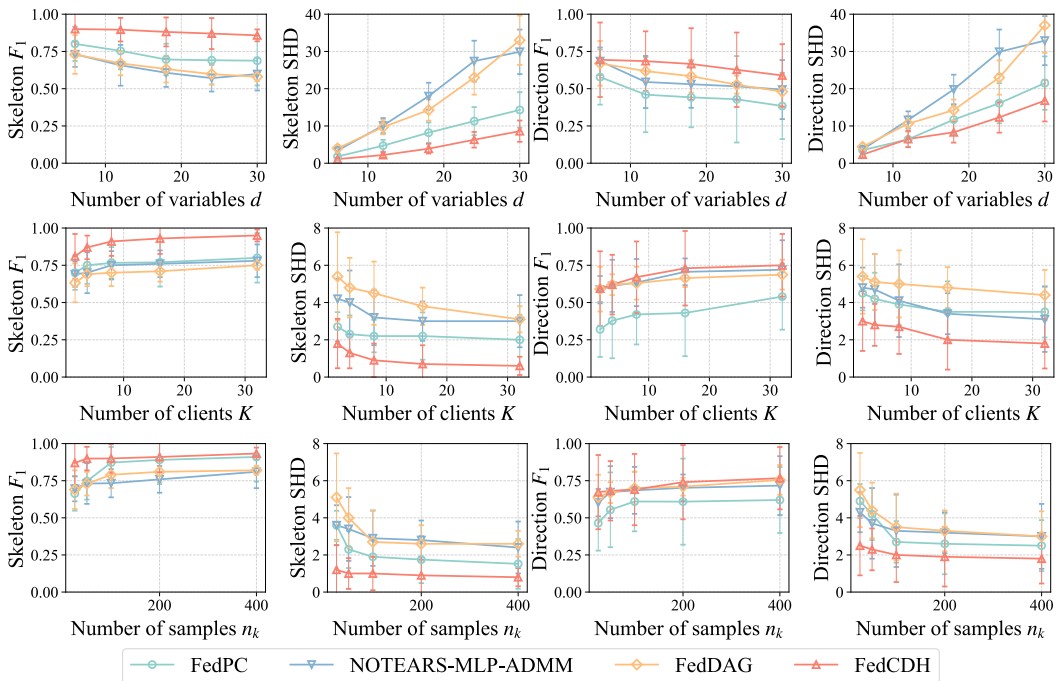

Figure A3: Results of synthetic dataset on general functional model. By rows, we evaluate varying number of variables $d$, varying number of clients $K$, and varying number of samples $n_k$. By columns, we evaluate Skeleton $F_1$ ($\uparrow$), Skeleton SHD ($\downarrow$), Direction $F_1$ ($\uparrow$) and Direction SHD ($\downarrow$).

### A7.3 RESULTS OF PRECISION AND RECALL

In the main paper, we have only provided the results of $F_1$ score and SHD, due to the space limit. Here, we provide more results and analysis of the precision and the recall. The results of average and standard deviation are exhibited in Figure A2. According to the results, we could observe that our FedCDH method generally outperformed all other baseline methods, regarding the precision of both skeleton and direction.

Moreover, in the linear Gaussian model, NOTEARS-ADMM generally achieved the best performance regarding the recall although it performed poorly in precision, the reason might be that NOTEARS-ADMM assumed homogeneous data distribution, which might face challenges in the scenarios with heterogeneous data. In the general functional model, when evaluating varying numbers of clients $K$ and samples $n_k$, FedDAG performed the best with respect to the recall, however, neither FedDAG nor NOTEARS-MLP-ADMM obtained satisfactory results in the precision, the reason might be that both of them are continuous-optimization-based methods, which might potentially suffer from various issues such as convergence and nonconvexity.

### A7.4 RESULTS OF COMPUTATIONAL TIME

Existing works about federated causal discovery rarely evaluate the computational time when conducting experiments. Actually, it is usually difficult to measure the exact computational time in real life, because of some facts, such as the paralleled computation for clients, the communication time costs between the clients and the server, and so on. However, the computational time is a significant factor to measure the effectiveness of a federated causal discovery method to be utilized in practical scenarios. Therefore, in this section, for making fair comparisons, we evaluate the computational time for each method, assuming that there is no paralleled computation (meaning that we record the computational time at each client and server and then simply add them up) and no extra communication cost (indicating zero time cost for communication).

We evaluate different settings as mentioned above, including varying number of variables $d$, varying number of clients $K$, and varying number of samples $n_k$. We generate data according to linear

Table A2: Results of computational time for varying number of variables $d$, varying number of clients $K$, and varying number of samples $n_k$. We report the average and standard deviation over 10 runs. This is the synthetic dataset based on linear Gaussian model.

| Data Sizes | | | Methods | | | |
|---|---|---|---|---|---|---|
| $d$ | $K$ | $n_k$ | FedPC | NOTEARS-ADMM | FedDAG | FedCDH (Ours) |
| 6 | | | $3.87 \pm 1.97$s | $14.10 \pm 1.89$s | $136.92 \pm 21.50$s | $8.14 \pm 2.47$s |
| 12 | | | $32.01 \pm 3.54$s | $28.33 \pm 2.46$s | $321.84 \pm 65.94$s | $62.69 \pm 7.77$s |
| 18 | 10 | 100 | $39.58 \pm 4.75$s | $35.13 \pm 2.89$s | $398.27 \pm 149.51$s | $98.57 \pm 9.23$s |
| 24 | | | $84.05 \pm 7.64$s | $40.01 \pm 2.94$s | $715.80 \pm 268.93$s | $172.11 \pm 18.18$s |
| 30 | | | $94.03 \pm 9.48$s | $56.35 \pm 3.91$s | $1441.13 \pm 519.04$s | $232.35 \pm 26.67$s |
| | 2 | | $0.72 \pm 0.24$s | $7.04 \pm 0.64$s | $50.38 \pm 11.29$s | $3.88 \pm 1.49$s |
| | 4 | | $2.07 \pm 0.73$s | $9.07 \pm 0.77$s | $85.08 \pm 15.68$s | $5.24 \pm 1.74$s |
| 6 | 8 | 100 | $3.64 \pm 1.54$s | $10.80 \pm 0.78$s | $114.81 \pm 29.67$s | $8.01 \pm 2.32$s |
| | 16 | | $5.79 \pm 2.59$s | $19.40 \pm 2.51$s | $342.34 \pm 62.28$s | $12.60 \pm 2.98$s |
| | 32 | | $14.08 \pm 4.44$s | $30.56 \pm 2.88$s | $714.06 \pm 137.31$s | $20.30 \pm 4.37$s |
| | | 25 | $0.48 \pm 0.10$s | $13.06 \pm 1.91$s | $125.77 \pm 20.64$s | $3.75 \pm 1.29$s |
| | | 50 | $1.47 \pm 0.64$s | $13.75 \pm 2.51$s | $127.25 \pm 20.38$s | $5.74 \pm 1.61$s |
| 6 | 10 | 100 | $3.87 \pm 1.97$s | $14.10 \pm 1.89$s | $136.92 \pm 21.50$s | $8.14 \pm 2.47$s |
| | | 200 | $16.52 \pm 3.63$s | $14.68 \pm 2.23$s | $138.67 \pm 31.91$s | $13.78 \pm 3.75$s |
| | | 400 | $51.10 \pm 6.87$s | $15.90 \pm 2.54$s | $140.37 \pm 34.42$s | $22.86 \pm 4.55$s |

Gaussian model. For each setting, we run 10 instances, report the average and the standard deviation of the computational time. The results are exhibited in Table A2.

According to the results, we could observe that among the four FCD methods, FedDAG is the least efficient method with the largest time cost, because it uses a two-level structure to handle the heterogeneous data: the first level learns the edges and directions of the graph and communicates with the server to get the model information from other clients, while the second level approximates the mechanism among variables and personally updates on its own data to accommodate the data heterogeneity. Meanwhile, FedPC, NOTEARS-ADMM and our FedCDH are comparable. In the setting of varying variables, our method exhibited unsatisfactory performance among the three methods, because the other two methods, FedPC and NOTEARS-ADMM, are mainly for homogeneous data. However, in the case of varying variables, NOTEARS-ADMM is the most ineffective method, because with the increasing of clients, more parameters (one client corresponds to one sub adjacency matrix which needs to be updated) should get involved in the optimization process, therefore, the total processing time can also increase by a large margin. In the scenario of varying samples, FedPC is the slowest one among the three methods.

## A7.5 HYPERPARAMETER STUDY

We conduct experiments on the hyperparameter, such as the number of mapping functions or hidden features $h$. Regarding the experiments in the main paper, we set $h$ to 5 by default. Here in this section, we set $h \in \{5, 10, 15, 20, 25, 30\}$, $d = 6$, $K = 10$, $n_k = 100$ and evaluate the performances. We generate data according to linear Gaussian model. We use the $F_1$ score, the precision, the recall and the SHD for both skeleton and direction. We also report the runtime. We run 10 instances and report the average values. The experimental results are given in Table A3.

According to the results, we could observe that with the number of hidden features $h$ increasing, the performance of the direction is obviously getting better, while the performance of the skeleton may fluctuate a little bit.

Theoretically, the more hidden features or a larger $h$ we consider, the better performance of how closely the random features approximate the kernels should be. When the number of hidden features approaches infinity, the performance of random features and that of kernels should be almost the same. And the empirical results seem to be consistent with the theory, where a large $h$ can lead to a higher $F_1$ score and precision for the directed graph.

Table A3: Hyperparameter study on the number of hidden features $h$. We evaluate the $F_1$ score, precision, recall, and SHD of both skeleton and direction. We report the average over 10 runs. This is the synthetic dataset based on linear Gaussian model.

| Metrics $h$ | Skeleton | | | | Direction | | | | Time↓ |
|---|---|---|---|---|---|---|---|---|---|
| | $F_1$ ↑ | Precision↑ | Recall↑ | SHD↓ | $F_1$ ↑ | Precision↑ | Recall↑ | SHD↓ | |
| 5 | 0.916 | 0.980 | 0.867 | 0.9 | 0.721 | 0.765 | 0.683 | 2.0 | 8.14s |
| 10 | 0.916 | 0.980 | 0.867 | 0.9 | 0.747 | 0.810 | 0.700 | 2.0 | 8.87s |
| 15 | 0.907 | 0.980 | 0.850 | 1.0 | 0.762 | 0.818 | 0.717 | 1.8 | 10.57s |
| 20 | 0.889 | 0.980 | 0.833 | 1.2 | 0.767 | 0.833 | 0.717 | 1.8 | 12.72s |
| 25 | 0.896 | 0.980 | 0.833 | 1.1 | 0.789 | 0.838 | 0.750 | 1.6 | 20.93s |
| 30 | 0.896 | 0.980 | 0.833 | 1.1 | 0.825 | 0.873 | 0.783 | 1.4 | 37.60s |

Table A4: Test result of statistical significance of our FedCDH method compared with other baseline methods. We report the p values via Wilcoxon signed-rank test (Woolson, 2007). This is the synthetic dataset based on linear Gaussian model.

| Parameters | | | [FedCDH vs. FedPC] | | | | [FedCDH vs. NOTEARS-ADMM] | | | | [FedCDH vs. FedDAG] | | | |
|---|---|---|---|---|---|---|---|---|---|---|---|---|---|---|
| d | k | n | S-$F_1$ | S-SHD | D-$F_1$ | D-SHD | S-$F_1$ | S-SHD | D-$F_1$ | D-SHD | S-$F_1$ | S-SHD | D-$F_1$ | D-SHD |
| 6 | 10 | 100 | 0.00 | 0.05 | 0.01 | 0.12 | 0.00 | 0.01 | 0.11 | 0.10 | 0.00 | 0.01 | 0.01 | 0.01 |
| 12 | 10 | 100 | 0.00 | 0.01 | 0.01 | 0.01 | 0.00 | 0.00 | 0.15 | 0.00 | 0.00 | 0.00 | 0.11 | 0.00 |
| 18 | 10 | 100 | 0.00 | 0.01 | 0.00 | 0.01 | 0.00 | 0.00 | 0.03 | 0.00 | 0.00 | 0.00 | 0.02 | 0.00 |
| 24 | 10 | 100 | 0.01 | 0.01 | 0.00 | 0.01 | 0.00 | 0.00 | 0.01 | 0.00 | 0.01 | 0.01 | 0.01 | 0.00 |
| 30 | 10 | 100 | 0.00 | 0.01 | 0.00 | 0.01 | 0.00 | 0.00 | 0.01 | 0.00 | 0.00 | 0.00 | 0.00 | 0.00 |
| 6 | 2 | 100 | 0.00 | 0.00 | 0.01 | 0.01 | 0.01 | 0.00 | 0.21 | 0.01 | 0.00 | 0.00 | 0.03 | 0.00 |
| 6 | 4 | 100 | 0.00 | 0.01 | 0.00 | 0.01 | 0.01 | 0.01 | 0.01 | 0.00 | 0.00 | 0.01 | 0.01 | 0.00 |
| 6 | 8 | 100 | 0.00 | 0.00 | 0.01 | 0.02 | 0.02 | 0.01 | 0.03 | 0.02 | 0.00 | 0.00 | 0.09 | 0.00 |
| 6 | 16 | 100 | 0.00 | 0.01 | 0.01 | 0.02 | 0.00 | 0.00 | 0.10 | 0.03 | 0.00 | 0.00 | 0.07 | 0.00 |
| 6 | 32 | 100 | 0.00 | 0.00 | 0.00 | 0.00 | 0.00 | 0.00 | 0.04 | 0.01 | 0.00 | 0.00 | 0.03 | 0.00 |
| 6 | 10 | 25 | 0.00 | 0.00 | 0.01 | 0.01 | 0.01 | 0.01 | 0.26 | 0.02 | 0.00 | 0.00 | 0.03 | 0.00 |
| 6 | 10 | 50 | 0.00 | 0.01 | 0.01 | 0.00 | 0.01 | 0.00 | 0.99 | 0.03 | 0.00 | 0.00 | 0.02 | 0.00 |
| 6 | 10 | 200 | 0.00 | 0.01 | 0.01 | 0.02 | 0.00 | 0.00 | 0.03 | 0.02 | 0.00 | 0.00 | 0.11 | 0.01 |
| 6 | 10 | 400 | 0.00 | 0.01 | 0.01 | 0.01 | 0.01 | 0.00 | 0.03 | 0.01 | 0.00 | 0.01 | 0.01 | 0.00 |

Moreover, the computational time is also increasing. When $h$ is smaller than 20, the runtime increases steadily. When $h$ is greater than 20, the runtime goes up rapidly. Importantly, we could see that even when $h$ is small, such as $h = 5$, the general performance of our method is still robust and competitive.

### A7.6 STATISTICAL SIGNIFICANCE TEST

In order to show the statistical significance of our method compared with other baseline methods on the synthetic linear Gaussian model, we report the p values via Wilcoxon signed-rank test (Woolson, 2007), as shown in Table A4. For each baseline method, we evaluate four criteria: Skeleton F1 (S-F1), Skeleton SHD (S-SHD), Direction F1 (D-F1), and Direction SHD (D-SHD).

We set the significance level to 0.05. Those p values higher than 0.05 are underlined. From the results, we can see that the improvements of our method are statistically significant at 5% significance level in general.

### A7.7 EVALUATION ON DENSE GRAPH

As shown in Figure 3 in the main paper, the true DAGs are simulated using the Erdös–Rényi model (Erdős et al., 1960) with the number of edges equal to the number of variables. Here we consider a more dense graph with the number of edges are two times the number of variables.

we evaluate on synthetic linear Gaussian model and general functional model, and record the $F_1$ score and SHD for both skeleton and directed graphs. All other settings are following the previous ones by default.

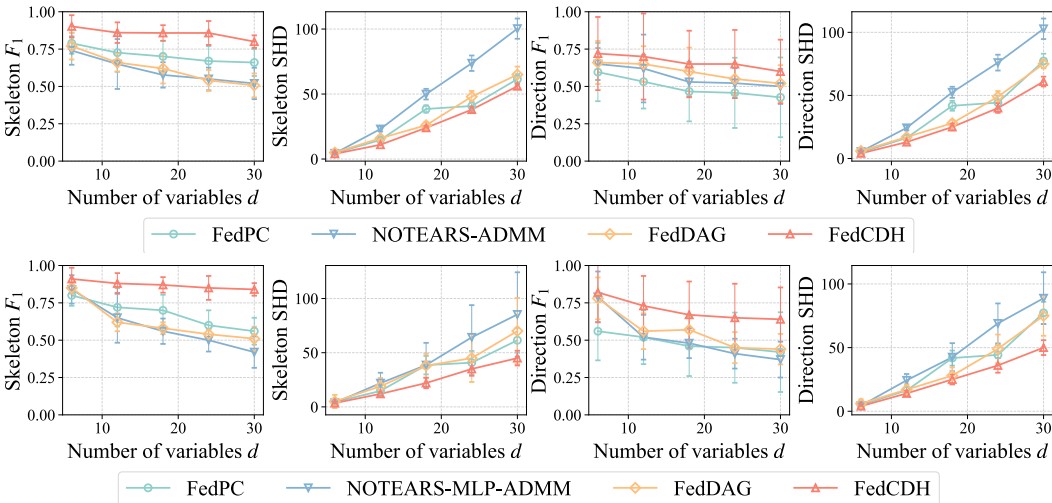

Figure A4: We evaluate on synthetic linear Gaussian model (Top Row) and general functional model (Bottom Row) when the number of edges are two times the number of variables. By columns, we evaluate Skeleton $F_1$ ($\uparrow$), Skeleton SHD ($\downarrow$), Direction $F_1$ ($\uparrow$) and Direction SHD ($\downarrow$).

According to the results as shown in Figure A4, we can see that our methods still outperformed other baselines in varying number of variables. Interestingly, when the generated graph is more dense, the performance of FedPC will obviously go down for various number of variables.

## A7.8 EVALUATION ON THE POWER OF CONDITIONAL INDEPENDENCE TEST

Here we added a new set of experiments to compare the power of our proposed federated conditional independence test and the centralized conditional independence test (i.e., kernel-based conditional independence test (Zhang et al., 2012)).

We followed the previous paper (Zhang et al., 2012) and used the post-nonlinear model (Zhang & Hyvarinen, 2012) to generate data. Assume there are four variables $W, X, Y$, and $Z$. $X = \hat{g}(\hat{f}(W) + \epsilon_X)$, $Y = \hat{g}(\hat{f}(W) + \epsilon_Y)$, and $Z$ is independent from both $X$ and $Y$. $\hat{f}$ and $\hat{g}$ are functions randomly chosen from linear, square, $\sin$ and $\tan$ functions. $\epsilon_X, \epsilon_Y, W$ and $Z$ are sampled from either uniform distribution $\mathcal{U}(-0.5, 0.5)$ or Gaussian distribution $\mathcal{N}(0, 1)$. $\epsilon_X$ and $\epsilon_Y$ are random noises. In this case, X and Y are dependent due to the shared component of $W$. Since $Z$ is independent from both $X$ and $Y$, therefore, we have $X \not\perp\!\!\!\perp Y | Z$. Here we set the significance level to 0.05, and the total sample size varies from 200, 400, 600, 800 to 1000. For federated CIT, we set the number of clients to 10, therefore, each client has 20, 40, 60, 80, or 100 samples. We run 1000 simulations and record the power of the two tests. From the result in Figure A5, we can see that the power of our federated CIT is almost similar to that of centralized CIT. Particularly, when the sample size reaches 1000, both of the two tests achieve power with more than 95%.

## A7.9 EVALUATION ON THE ORDER OF DOMAIN INDICES

In this section, we aim to find out whether the order of domain indices will impact the results. Theoretically, there should be no impact on the results when it takes different values because this domain index $\mho$ is essentially a discrete variable (more specifically, a categorical variable, with no numerical order among different values), a common approach to deal with such discrete variable is to use delta kernel (based on Kronecker delta function), and therefore it is reasonable to use random features to approximate the delta kernel for discrete variables.

Empirically, we have added one new set of experiments to evaluate whether the order of domain indices will impact the results. We have one set of domain indices and run our FedCDH on the synthetic linear Gaussian model with varying number variables $d \in \{6, 12, 18, 24, 30\}$ while keeping $K = 10$ and $n_k = 100$, other settings are the same as those in our main paper. Then, we randomly shuffle the indices for different domains, denoted by "FedCDH+Shuffle".

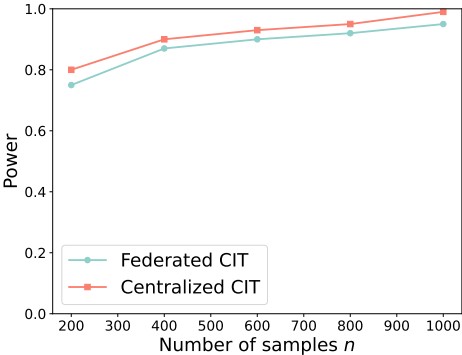

Figure A5: Comparison regarding the power of test between federate conditional independence test and the centralized conditional independence test.

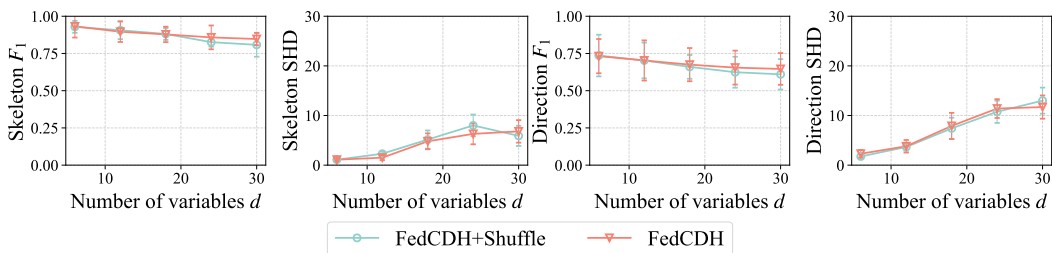

Figure A6: Evaluation on the order of domain indices on linear Gaussian model. We evaluate varying number of variables $d$. By columns, we evaluate Skeleton $F_1$ ($\uparrow$), Skeleton SHD ($\downarrow$), Direction $F_1$ ($\uparrow$) and Direction SHD ($\downarrow$).

As shown in Figure A6, the results turned out that: the performances between the two sets of different domain indices are quite similar, and we may conclude that it has no obvious impact on the results when the domain indices take different values.

## A8    Details about the Experiments on Real-world Dataset

### A8.1    Details about fMRI Hippocampus Dataset

We evaluate our method and the baselines on fMRI Hippocampus (Poldrack et al., 2015). The directions of anatomical ground truth are: PHC $\to$ ERC, PRC $\to$ ERC, ERC $\to$ DG, DG $\to$ CA1, CA1 $\to$ Sub, Sub $\to$ ERC and ERC $\to$ CA1. Generally, we follow a similar setting as the experiments on synthetic datasets. For each of them, we use the structural Hamming distance (SHD), the $F_1$ score as evaluation criteria. We measure both the undirected skeleton and the directed graph. Here, we consider varying number of clients $K$ and varying number of samples in each client $n_k$.

The results of $F_1$ score and SHD is given in Figure A7. According to the results, we could observe that our FedCDH method generally outperformed all other baseline methods, across all the criteria listed. The reason could be that our method is specifically designed for heterogeneous data while some baseline methods assume homogeneity like FedPC and NOTEARS-MLP-ADMM, furthermore, our method can handle arbitrary functional causal models, different from some baseline methods that assume linearity such as FedPC. Compared with our method, FedDAG performed much worse, the reason might be its nature of the continuous optimization, which might suffer from various issues such as convergence and nonconvexity.

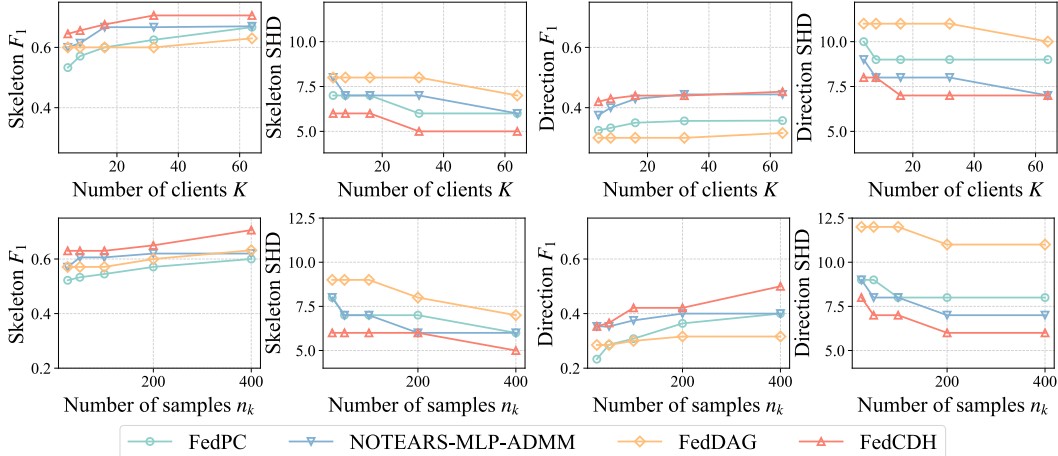

Figure A7: Results of real-world dataset fMRI Hippocampus (Poldrack et al., 2015). By rows, we evaluate varying number of clients $K$ and varying number of samples $n_k$. By columns, we evaluate Skeleton $F_1$ (↑), Skeleton SHD (↓), Direction $F_1$ (↑) and Direction SHD (↓).

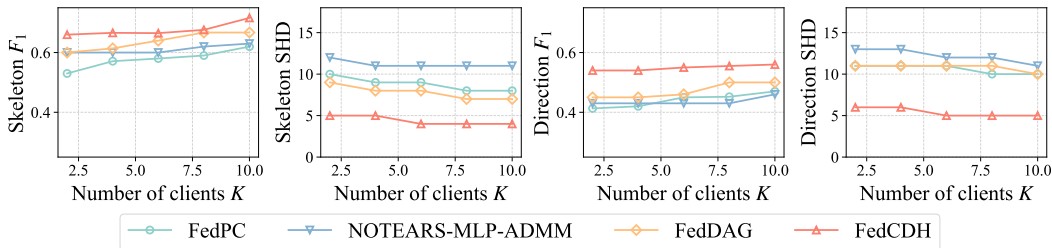

Figure A8: Results of real-world dataset HK Stock Market (Huang et al., 2020). We evaluate varying number of clients $K$, and we evaluate Skeleton $F_1$ (↑), Skeleton SHD (↓), Direction $F_1$ (↑) and Direction SHD (↓).

## A8.2  DETAILS ABOUT HK STOCK MARKET DATASET

We also evaluate on HK stock market dataset (Huang et al., 2020) (See Page 41 for more details about the dataset). The HK stock dataset contains 10 major stocks, which are daily closing prices from 10/09/2006 to 08/09/2010. The 10 stocks are Cheung Kong Holdings (1), Wharf (Holdings) Limited (2), HSBC Holdings plc (3), Hong Kong Electric Holdings Limited (4), Hang Seng Bank Ltd (5), Henderson Land Development Co. Limited (6), Sun Hung Kai Properties Limited (7), Swire Group (8), Cathay Pacific Airways Ltd (9), and Bank of China Hong Kong (Holdings) Ltd (10). Among these stocks, 3, 5, and 10 belong to Hang Seng Finance Sub-index (HSF), 1, 8, and 9 belong to Hang Seng Commerce and Industry Sub-index (HSC), 2, 6, and 7 belong to Hang Seng Properties Sub-index (HSP), and 4 belongs to Hang Seng Utilities Sub-index (HSU).

Here one day can be also seen as one domain. We set the number of clients to be $K \in \{2, 4, 6, 8, 10\}$ while randomly select $n_k = 100$ samples for each client. All other settings are following previous ones by default. The results are provided in Figure A8. According to the results, we can infer that our FedCDH method also outperformed the other baseline methods, across the different criteria. Similar to the analysis above, our method is tailored for heterogeneous data, in contrast to baseline methods like FedPC and NOTEARS-MLP-ADMM, which assume homogeneity. Additionally, our approach is capable of handling arbitrary functional causal models, setting it apart from baseline methods like FedPC that assume linearity. When compared to our method, FedDAG exhibited significantly poorer performance. This could be attributed to its reliance on continuous optimization, which may encounter challenges such as convergence and nonconvexity.

