# OpenReview forum: "Federated Causal Discovery from Heterogeneous Data"
_ICLR.cc/2024/Conference — ICLR 2024 poster_

### Official Review · Reviewer_VxyA · 2023-10-28

**Soundness:** 3 good
**Presentation:** 1 poor
**Contribution:** 3 good
**Rating:** 5
**Confidence:** 4

**Summary:**

This paper focuses on the problem of federated causal discovery from heterogeneous data, and proposes a novel constraint-based method called FedCDH, which can accommodate arbitrary causal models and heterogeneous data. It constructs the summary statistics to protect data privacy and further proposes federated conditional independence test (FCIT) and federated independent change principle (FICP) for skeleton discovery and direction determination. The experimental results on synthetic and real datasets show the efficacy of the proposed method.

**Strengths:**

1. The paper addresses the issue of heterogeneous data in federated causal discovery and relaxes the assumptions of causal models, which are critical problems.
2. The paper proposes a novel constraint-based method for effectively conducting federated causal discovery from heterogeneous data.
3. The paper provides detailed proofs for the presented theorems and lemmas. Extensive experiments demonstrate the effectiveness of the proposed method.

**Weaknesses:**

1. The paper conducts numerous experiments; however, it should provide a more in-depth analysis of the underlying reasons behind the experimental results, rather than just stating the observations.
2. The section of method overuses symbols, leading to difficulties in understanding.
3. The presentation of this paper can be largely improved for clarification.

**Questions:**

1. In the penultimate paragraph of page 2, the authors say ‘Let k be the client index, and ℧ be the domain index’, what’s the difference between client and domain?
2. In the last paragraph of page 2, the authors say ‘When the data is heterogeneous, there must be some causal models changing across different domains. The changes may be caused by the variation of causal strengths or noise variances.’ The authors should better clarify what is the change of causal models.
3. In the first paragraph of page 3, ψ(℧) and θi(℧) are functions of ℧, and ℧ is a positive integer from 1 to k. Actually, ℧ is a value defined by the authors, and does it have an impact on the results when it takes different values?
4. In the fourth paragraph of page 3, as indicated by the authors, ℧ and Vi are connected by unobserved domain-changing variables ψ(℧) and θi(℧), so what does it mean of ‘If there is an edge between surrogate variable ℧ and observed variable Vi on Gaug’?
5. The authors should improve the presentation quality of the paper and fix typos. For example:
(1) In the second paragraph of page 5, ‘therefore, we would like to …’ -> ‘Therefore, we would like to …’.
(2) In the second paragraph of the Section of A6.4 Results of Computational Time, ‘The results are exhibited in Table A3.’ -> ‘The results are exhibited in Table A1’.

---

> ### Author Response · Authors · 2023-11-17
> **Responses to Reviewer VxyA (1/2)**
>
> We appreciate the reviewer for the time dedicated to reviewing our paper, constructive suggestions, and encouraging feedback. We have carefully **modified the manuscript** in light of your detailed suggestions. Particularly, in order to distinguish between different symbols and avoid confusion, we have **added a new notation table** in Appendix A1. Moreover, **a new set of experiments** is added to evaluate the impact on domain indices. Please find the responses to all your comments below.
>
>
> **Q1:** "The paper conducts numerous experiments; however, it should provide a more in-depth analysis of the underlying reasons behind the experimental results, rather than just stating the observations."
>
> **A1**: Thanks a lot for the constructive suggestion. We would like to clarify that: due to the space limit in the main paper, we have provided some of the detailed analysis in the Appendix. For example, in Appendix A7.2, we concluded the observational results w.r.t. $F_1$ and SHD on synthetic datasets and then provided in-depth analysis on why those phenomena might happen (e.g., FedPC performed the worst, due to its strong assumptions on linear model and homogeneous data; FedDAG and NOTEARS-MLP-ADMM presented worse SHD results compared to our FedCDH, probably because both of them are continuous-optimization-based methods, which might suffer from various issues such as convergence and nonconvexity).
>
> Certainly, we are glad to incorporate your suggestion, to provide a more in-depth analysis for all other experiments. Therefore, we have updated our manuscript with a more detailed analysis, as you can see in Appendix A7.3, A7.4, A7.5, A8.1, and A8.2.
>
>
> **Q2:** "The section of method overuses symbols, leading to difficulties in understanding."
>
> **A2**: We are grateful for your straightforward and helpful comment. In order to improve readability and avoid symbol confusion, we have summarized and categorized the symbols throughout the paper and put them into the table in Appendix A1. Hopefully, this table can be helpful for the readers, especially when reading the equations and theorems.
>
>
> **Q3:** "The presentation of this paper can be largely improved for clarification."
>
> **A3**: Thank you so much for the comment. We have tried our best to improve the presentation based on your comments. Hopefully, our responses to your detailed question below from Q4 to Q8 and the accordingly modified manuscript can help to improve our presentation. If there are any other comments you have on the presentation, we would appreciate it if you could kindly let us know.
>
>
> **Q4:** "In the penultimate paragraph of page 2, the authors say ‘Let k be the client index, and $\mho$ be the domain index’, what’s the difference between client and domain?"
>
> **A4:** Thanks a lot for this great question. A client is a decentralized entity in federated learning, while a domain typically refers to a distinct category of data that share common features. Usually, when one set of data is heterogeneous, we say that the data distribution is changing across different domains. In our paper, we assume that one client corresponds to one unique domain. We have updated this detail in Section 2 of our manuscript.
>
>
> **Q5:** "In the last paragraph of page 2, the authors say ‘When the data is heterogeneous, there must be some causal models changing across different domains. The changes may be caused by the variation of causal strengths or noise variances.’ The authors should better clarify what is the change in causal models.
>
> **A5:** Thanks for raising this great suggestion. By "the change of causal model", we intend to say that: the structural causal model (SCM) $V_i = f_i(PA_i, \epsilon_i)$ is different/changing across different domains, that is, the function $f_i$ or the distribution of the noise $\epsilon_i$ is different/changing across different domains.
>
> Taking a linear SCM as an example, more specifically, we have $V_i = \sum_{j\in PA_i} a_{ij} V_j + \epsilon_i$, where $a_{ij}$ are the coefficients or causal strengths in the causal function and $\epsilon_i$ is the noise term. When we say the causal models change, then either the causal strength $a_{ij}$ or the noise variance of $\epsilon_i$ will change across domains.
>
> We have included the discussion in Section 2 of the updated manuscript.

---

> > ### Author Response · Authors · 2023-11-17
> > **Responses to Reviewer VxyA (2/2)**
> >
> > **Q6:**  "In the first paragraph of page 3, $\psi({\mho})$ and $\theta_i({\mho})$ are functions of $\mho$, and $\mho$ is a positive integer from 1 to K. Actually, $\mho$ is a value defined by the authors, and does it have an impact on the results when it takes different values?"
> >
> > **A6:** Thanks a lot for this insightful question. Theoretically, there is no impact on the results when it takes different values because this surrogate variable $\mho$ is essentially a discrete variable (more specifically, a categorical variable, with no numerical order among different values), a common approach to deal with such discrete variable is to use delta kernel (based on Kronecker delta function), and therefore it is reasonable to use random features to approximate the delta kernel for discrete variables. We have included these discussions in Section 3.3 of the updated manuscript.
> >
> > Empirically, we have added one new set of experiments in Appendix A7.9 where we randomly shuffle the domain indices, and the results show that: the performances between the two sets of different domain indices are quite similar, and we may conclude that it has no obvious impact on the results when the domain indices take different values.
> >
> > **Q7:** "In the fourth paragraph of page 3, as indicated by the authors, $\mho$ and $V_i$ are connected by unobserved domain-changing variables $\psi(\mho)$ and $\theta_i(\mho)$, so what does it mean of ‘If there is an edge between surrogate variable $\mho$ and observed variable $V_i$ on $\mathcal{G}_{aug}$’?"
> >
> > **A7:** Thank you so much for this great point. In the context of the simplified augmented graph, for example in Figure 1(b), if there is an edge pointing from the surrogate variable $\mho$ to the observed variable $V_i$, we can say that the causal model related to $V_i$ is changing across different domains, in other words, the data distribution of $V_i$ is heterogeneous across domains. On the contrary, if there is no edge between the surrogate variable $\mho$ to the observed variable $V_k$, we can say that the data distribution of $V_k$ is invariant or homogeneous across domains. We have updated those details in Section 2 of our manuscript.
> >
> > **Q8:** The authors should improve the presentation quality of the paper and fix typos. For example: (1) In the second paragraph of page 5, ‘therefore, we would like to ...’ -> ‘Therefore, we would like to ...’. (2) In the second paragraph of the Section of A6.4 Results of Computational Time, ‘The results are exhibited in Table A3.’ -> ‘The results are exhibited in Table A1’.
> >
> > **A8:** Thanks a lot for pointing them out, which indeed helps to improve the presentation of the manuscript. We have corrected them accordingly and re-checked the manuscript thoroughly.
> >
> >
> > Thank you again for your constructive comments, which are really helpful in improving the quality of the manuscript. We hope our responses, the new experiments, and the modified manuscript can adequately address the remaining concerns. Please let us know if there are any further questions.

---

> > > ### Comment · Reviewer_VxyA · 2023-11-20
> > >
> > > Thank you for your clear response. Most of my concerns have been addressed and I have also read other comments and detailed responses. However, I still have concerns regarding the presentation of this paper.

---

> ### Author Response · Authors · 2023-11-20
> **Could you please let us know more details about your concerns?**
>
> Dear Reviewer VxyA:
>
> We are very happy that most of your concerns have been well addressed. Thanks for letting us know that you still have concerns regarding the presentation. Could you please kindly let us know exactly what presentation issue you saw with the paper (e.g., which Section or which Figure), so that we can address them as soon as possible?
>
> Sincerely,
>
> Authors of Submission 3985.

---

### Official Review · Reviewer_157B · 2023-10-30

**Soundness:** 3 good
**Presentation:** 2 fair
**Contribution:** 3 good
**Rating:** 5
**Confidence:** 4

**Summary:**

In this paper, the authors propose a novel FCD method attempting to accommodate arbitrary causal models and heterogeneous data. Specifically, they first utilize a surrogate variable corresponding to the client index to account for the data heterogeneity across different clients. Then they develop a federated conditional independence test (FCIT) for causal skeleton discovery and establish a federated independent change principle (FICP) to determine causal directions. These approaches involve constructing summary statistics
as a proxy of the raw data to protect data privacy. Owing to the nonparametric properties, FCIT and FICP make no assumption about particular functional forms, thereby facilitating the handling of arbitrary causal models. Extensive experiments on synthetic and real datasets could show the efficacy of the proposed method.

**Strengths:**

This paper features a detailed theoretical analysis and addresses a highly challenging problem.

**Weaknesses:**

This paper is relatively difficult to read, and it is not very comprehensible, making it not conducive for others to follow and reproduce the work. This paper presents a list of theorems, but I haven't seen a clear explanation of the specific challenging problem you have addressed. The relationship between these theorems and the contributions of the paper is quite ambiguous.

**Questions:**

Please see the weakness.

---

> ### Author Response · Authors · 2023-11-17
> **Responses to Reviewer 157B (1/2)**
>
> We appreciate the reviewer for the time dedicated to reviewing our paper and constructive suggestions. In light of your valuable feedback, we have **updated our manuscript** in the first paragraph of Section 3. To improve the readability of our paper, we have **added a new notation table** in Appendix A1. Please find our responses to your concerns below.
>
>
> **Q1:** "This paper is relatively difficult to read, and it is not very comprehensible, making it not conducive for others to follow and reproduce the work. This paper presents a list of theorems, but I haven't seen a clear explanation of the specific challenging problem you have addressed. The relationship between these theorems and the contributions of the paper is quite ambiguous."
>
> **A1:** Thanks for raising those concerns, which will help to improve the presentation of our manuscript. First of all, in order to improve the readability of our paper, we have summarized the symbols and put them in the table in Appendix A1. Then, to address your main concern, we would like to clarify two aspects: what the contributions of our paper are, and how those theorems and lemmas in the paper are related to each other. (The key theorems, in which our main contributions lie, are in **bold**.)
>
> **The main contributions and the challenging problem we have addressed:**
> - First, in this paper we proposed a novel constraint-based method for federated causal discovery (FCD), which can handle arbitrary functional causal models and heterogeneous data, demonstrating broader applicability compared to the existing FCD methods.
> - Specifically, we proposed two novel submodules, i.e., federated conditional independence test (FCIT; **Theorem 4 and Theorem 5**) and federated independent change principal (FICP; **Theorem 6**), for skeleton discovery and direction determination. Intuitively, they are the federated extensions of previous centralized conditional independent test (CIT) [1] and independent change principal (ICP) [2], respectively. To the best of our knowledge, we are the first work to propose the methods for federated CIT and federated ICP.
>     - In order to satisfy the demand that the raw data must not be directly shared across domains (to protect the data privacy), we proposed summary statistics (**Theorem 8**), consisting of total sample size and covariance tensor. Fortunately, our summary statistics are sufficient to represent all the statistics that appeared in FCIT and FICP.
> - Lastly, to evaluate the efficacy of our proposed method, we conducted extensive experiments on two synthetic datasets (i.e., linear Gaussian model and general functional model) and another two real-world datasets (i.e., fMRI Hippocampus dataset and HK stock market dataset).
>
> **The relationships among the theorems/lemmas:**
>
> In short, Lemma 1 and 2 are the preliminaries for our method; Lemma 3 is an extension of Lemma 1; Theorem 4 and Theorem 5 are the core parts of FCIT whose proofs rely on Lemma 3; Theorem 6 is the core part of FICP which is based on Lemma 2; Lemma 7 shows the main rules to help derive Theorem 4, 5 and 6; And Theorem 8 concludes that what summary statistics are and how summary statistics are sufficient to achieve FCD.
>
> - Lemma 1 and Lemma 2 provide the fundamental characterizations for the centralized CIT and ICP, respectively. Based on those two lemmas, we try to extend them from centralized versions to the corresponding federated versions.
> - Lemma 3 is an extension of Lemma 1, presenting another characterization of conditional independence (CI). With Lemma 3, we can measure the CI using covariance matrices.
> - **Theorem 4** exhibits the core of our proposed FCIT, with the new test statistics and the asymptotic distribution.
> - **Theorem 5** shows how we can approximate the asymptotic distribution with a two-parameter Gamma distribution for the test, where the mean and the variance are displayed.
> - **Theorem 6** presents our proposed statistics for FICP. Fortunately, the test statistic in Theorem 4, the mean and variance in Theorem 5, and the dependence values in Theorem 6 are all represented in relation to the variance or covariance matrices, which are decomposable with respect to the samples.
> - Lemma 7 is the connection rule that builds the relationship between the kernel matrix and covariance matrix. The rules in Lemma 7 actually help to derive the proofs of Theorem 4, 5, and 6.
> - **Theorem 8** states the sufficiency of our proposed summary statistics. With merely these decomposable summary statistics, we can complete the whole FCD process.
>
>
> We sincerely thank you once again for the valuable and constructive suggestions. We hope you will find that our responses, along with updated manuscripts, have properly addressed your concerns. Please kindly let us know if there are any further questions or comments.

---

> > ### Author Response · Authors · 2023-11-17
> > **Responses to Reviewer 157B (2/2)**
> >
> > **References:**
> >
> > [1] Zhang, Peters, et al. "Kernel-based conditional independence test and application in causal discovery." UAI, 2012.
> >
> > [2] Huang, Zhang, et al. "Causal discovery from heterogeneous/nonstationary data."  Journal of Machine Learning Research, 2020.

---

> ### Author Response · Authors · 2023-11-22
> **Could you please kindly check whether our responses properly addressed your concerns?**
>
> Dear Reviewer 157B:
>
> Thank you for your time and efforts in reviewing our paper. We have provided responses and revised the paper according to your constructive comments.
>
> As the rolling discussion period for our paper is coming to a close, we are still waiting for your feedback on our responses. Could you please kindly check whether our responses properly addressed your concerns? If there are any other concerns, please kindly let us know, so that we can address them as soon as possible. Thank you so much!
>
> Sincerely,
> Authors

---

### Official Review · Reviewer_SgBE · 2023-10-31

**Soundness:** 4 excellent
**Presentation:** 3 good
**Contribution:** 3 good
**Rating:** 8
**Confidence:** 3

**Summary:**

The paper proposes a constraint-based federated causal discovery method suitable for heterogeneous data. That is, the paper assumes different but related structural causal models at each client and seeks to discover the global structural causal model where local differences are modeled using domain-specific effective parameters for each variable and a set of domain-specific the pseudo-confounders. The paper proposes to use the Hilbert-Schmidt independence criterion and a (federated) independent change principle, based on the partial cross-covariance operator on a reproducing kernel Hilbert space, which captures non-linearities in the relationship of variables based on the chosen kernel. To make computations tractable, it approximates the kernel with random Fourier features. It shows that these criteria can be computed from summary statistics based on the kernel matrices.

**Strengths:**

- federated causal discovery is a relevant and interesting problem
- the method is well motivated and theoretically sound
- good empirical performance

**Weaknesses:**

- the method requires that the different domains are known in the application

**Questions:**

**Questions:**
- why assume $L$ hidden confounders? How is L set in practice? Could we simply choose $d$ or $2^d$? In the theoretical evaluation, $L$ is assumed to be equal to the number of datapoints n which goes to infinity. Isn't that a problematic assumption, or did I misread the proof of Thm. 4?
- How does sharing local kernel matrices affect privacy?

**Detailed Comments:**
- The notation with bold $\mathbf{\psi}$ and normal $\psi$ is visually hard to distinguish.
- The formulation with an underlying causal graph and an augmented causal graph is not clear to me. For me it made sense to either assume an underlying "augmented" graph that captures local differences, or to assume an underlying graph and local interventions - these interventions are then modeled via the augmented graph. If the latter is the case, this should be clarified.
- Is it necessary to consider $\mho$ an observable variable? In the experiments, some additional information is used to decide the domain, but in general this is rarely possible. E.g., in [1] the domain is infered. What happens if you assumed that every client has a different domain and $\mho$ is the client index?


[1] Mian, Osman, Michael Kamp, and Jilles Vreeken. "Information-theoretic causal discovery and intervention detection over multiple environments." Proceedings of the AAAI Conference on Artificial Intelligence, AAAI-23. 2023.

---

> ### Author Response · Authors · 2023-11-17
> **Responses to Reviewer SgBE**
>
> We appreciate the reviewer for the time dedicated to reviewing our paper, constructive suggestions, and encouraging feedback. We have carefully **modified the manuscript** in light of your detailed suggestions. Particularly, in order to distinguish between different symbols and avoid confusion, we have **added a symbol table** in Appendix A1. Please find the responses to all your comments below.
>
> **Q1:** "Why assume $L$ hidden confounders? How is $L$ set in practice? Could we simply choose $d$ or $2^d$? In the theoretical evaluation, $L$ is assumed to be equal to the number of datapoints $n$ which goes to infinity. Isn't that a problematic assumption, or did I misread the proof of Theorem 4?"
>
> **A1:** Thanks for raising this great question. In Section 2, we use $L$ to denote the number of hidden confounders, where $L$ can be varying. The minimum value for $L$ can be 0, meaning that there is no latent confounder in the graph, while the maximum value can be $C_d^2=\frac{d(d+1)}{2}$ where $d$ is the number of observed variables, meaning that each pair of observed variables has a hidden confounder. Fortunately, we can directly work on the simplified augmented graph $\mathcal{G}_{aug}$ with surrogate variable $\mho$ as shown in Figure 1(b), where we have no need to specify the value of $L$. We have updated this discussion in Section 2 of our manuscript.
>
> However, $L$ in the proof of Theorem 4, has a different meaning, denoting the number of nonzero eigenvalues of the kernel matrices. We apologize for the double use of one symbol. We have replaced $L$ with $\beta$, and updated the proof of Theorem 4 in our manuscript.
>
> Furthermore, in order to improve the readability of our paper, we have summarized the most important symbols throughout the paper and added the notation table in Appendix A1.
>
> **Q2:** "How does sharing local kernel matrices affect privacy?"
>
> **A2:** Thanks for the question. We would like to clarify that: our method does not share the kernel matrices, instead, it shares our constructed covariance tensors. As we mentioned in Section 3.4, sharing the covariance tensor instead of the raw data can preserve data privacy to some extent. Moreover, if each client is required to not directly share the local covariance tensor, one can use some secure computation techniques such as secure multiparty computation [2], to further enhance the protection of data privacy.
>
> **Q3:** "The notation with bold $\boldsymbol{\psi}$ and normal $\psi$ is visually hard to distinguish."
>
> **A3:** Thanks for the great suggestions, which help improve the clarity of our manuscript. In order to distinguish between the two symbols, we have replaced the bold one $\boldsymbol{\psi}$ with $\boldsymbol{\tilde{\psi}}$. We have updated the manuscript accordingly in Section 2.
>
> **Q4:** "The formulation with an underlying causal graph and an augmented causal graph is not clear to me. For me it made sense to either assume an underlying "augmented" graph that captures local differences or to assume an underlying graph and local interventions - these interventions are then modeled via the augmented graph. If the latter is the case, this should be clarified."
>
> **A4:** Thanks for the constructive suggestion. We agree with the latter point that we assume an underlying graph and local interventions, which are then modeled via the augmented graph. In light of your suggestion, we have updated the manuscript accordingly.
>
> **Q5:** "Is it necessary to consider $\mho$ an observable variable? In the experiments, some additional information is used to $\mho$ decide the domain, but in general this is rarely possible. E.g., in [1] the domain is inferred. What happens if you assume that every client has a different domain and $\mho$ is the client index?"
>
> **A5:** We appreciate your insightful comment. That is exactly our assumption—each client has a different domain and $\mho$ is the client/domain index (one client corresponds to one domain). Such an assumption is natural in the federated learning setting, because the data distribution may often be heterogenous across different clients/domains. Therefore, this client/domain index is an observable variable. We will append this detail in our revised manuscript.
>
> Thank you again for your constructive comments, which are really helpful in improving the quality of the manuscript. Meanwhile, thanks a lot for the appreciation of our work, and we are really encouraged by it. We hope our responses and the modified manuscript can adequately address the remaining concerns. Please let us know if there are any further questions.
>
>
> ---
> **References:**
>
> [1] Mian, Kamp, and Vreeken. "Information-theoretic causal discovery and intervention detection over multiple environments." AAAI, 2023.
>
> [2] Cramer, Damgard, et al. "Secure multiparty computation." Cambridge University Press, 2015.

---

> > ### Comment · Reviewer_SgBE · 2023-11-21
> > **Response to authors**
> >
> > Thank you for your detailed response. You have answered my concerns and I maintain my positive assessment. Out of curiosity: you encode the client index with $\mho$ under the assumption that each client has a different domain. In practice, many clients might share the same domain, though. Wouldn't that lead to a large number of superfluous hidden confounders? Would it be possible to infer which domains are similar? Or maybe even which confounder variables essentially encode the same intervention?

---

> > > ### Author Response · Authors · 2023-11-21
> > > **Response to Reviewer SgBE**
> > >
> > > We are very happy that your concerns have been well addressed. Thanks for letting us know there are some questions that you are still curious about. Our responses to these questions are given below.
> > >
> > > Since our paper mainly focuses on heterogeneous data, that is why we assume each client has a different domain. However, our method can also be applied to the special case where all domains are the same (i.e., the data is homogeneous in all domains). In this case, there will be no edge to be found in our estimated causal graph between the surrogate variable (or domain index) $\mho$ and the observed variables $\boldsymbol{V}$, because no matter how $\mho$ changes, the distribution of $\boldsymbol{V}$ will always be the same, meaning that $\mho$ and $\boldsymbol{V}$ are independent. Furthermore, we can also leverage this characterization to infer which domains are similar or the same: feed the data from the interested domains to our method and check the estimated graph, if there is no edge between the the surrogate variable and all the observed variables, we can conclude that these domains are the same.
> > >
> > > Please let us know if there are any further questions. Thank you!

---

### Official Review · Reviewer_j4X7 · 2023-11-06

**Soundness:** 3 good
**Presentation:** 3 good
**Contribution:** 2 fair
**Rating:** 5
**Confidence:** 4

**Summary:**

This work addresses the problem of federated causal discovery, where one wishes to recover the causal structure of some domain given datasets from multiple independent sources. The authors address this problem by proposing a procedure that leverages summary statistics, rather than individual observations, from each source. The work proposes KCIT, and in order to allow for the adaptation to the federated setting random fourier features are used for an approximation. The work also proposes a federated version of the individual change principle, also leveraging summary statistics. Empirical results are provided which show favorable performance.

**Strengths:**

This paper proposes a very sensible extension to existing work on conditional independence testing and the invariance condition to the federated learning setting. The authors lay out the proposed method well, and the approach is easy to follow and understand. In addition, the experimental results show nice performance in comparison to other methods.

**Weaknesses:**

In my view the largest issue with this method is that there are a few places that lack sufficient specificity. The authors use the kernel conditional independence test as a basis, but don't appear to specify the necessary assumptions on the underlying functional forms, same for the invariance principle. It also isn't clear to me how we should expect the behavior of the federated approach to compare to its non-federate counterpart, e.g., what assumptions are necessary on the number of samples per domain? What is loss of power between the federate and non-federated tests? The authors appeal to random fourier features, which seems necessary (or at least some approximation appears to be necessary) but it isn't clear to me under which conditions we should expect this algorithm to not pay a price for this approximation. I certainly could be missing something, but the proofs provided appear to rely on large sample properties, but it's really unclear to me when those start to kick in, e.g. when should we expect the test to converge? How many samples before the summary statistics applied to the random Fourier features become a reliable representation of the underlying distribution for a given source?

**Questions:**

All of my questions are largely laid out above. In general, it would be good to get a sense of the finite sample behavior of the proposed method.

---

> ### Author Response · Authors · 2023-11-17
> **Responses to Reviewer j4X7 (1/2)**
>
> We appreciate the reviewer for the time dedicated to reviewing our paper and constructive suggestions. With the help of this valuable feedback, we believe that our manuscript could be improved a lot. We have added **a new set of experiments** for evaluating the power of the test, and **updated our manuscript** according to your detailed suggestions. Please find the point-by-point responses below.
>
> **Q1**: "The authors use the kernel conditional independence test as a basis, but don't appear to specify the necessary assumptions on the underlying functional forms, same for the invariance principle."
>
> **A1**: We appreciate this insightful comment. Fortunately, these kernel-based methods do not make restricted or specific assumptions on the underlying functional causal models due to the nonparametric properties, in other words, they can handle both linear or nonlinear functions, Gaussian or non-Gaussian data distributions well.
>
> However, once the Gaussian kernel is utilized, the kernel conditional independence test implicitly assumes smoothness for the relationship of continuous variables. Similarly, the smoothness assumption is also used for the invariance principle. We have included the details at the end of Section 2 of our manuscript.
>
>
> **Q2**: “It also isn't clear to me how we should expect the behavior of the federated approach to compare to its non-federate counterpart, e.g., what assumptions are necessary on the number of samples per domain? What is loss of power between the federate and non-federated tests?“
>
> **A2**: Thanks a lot for the insightful questions. We would like to address your concerns one by one as follows.
> - Ideally speaking, the performance of our federated approach should be approaching that of its non-federated counterpart [1].
>
> - Theoretically, the number of samples should be approaching infinity in order to satisfy the guarantee, e.g., Equation (31) in the proof of Theorem 4 in Appendix A4.2. However, in practice, based on Figure 3 and Figure 4, we found that at least 100 samples per domain can lead to a competitive performance.
>
> - Regarding the loss of power between the federated tests and non-federated tests, we have added the new experiment results in Appendix A7.8. Specifically, we found that the performance of the federated test is quite similar to that of the centralized test [1], and the loss of power is quite small.
>
>
> **Q3**: "The authors appeal to random Fourier features, which seems necessary (or at least some approximation appears to be necessary) but it isn't clear to me under which conditions we should expect this algorithm to not pay a price for this approximation. "
>
> **A3**: Thanks for raising this great point. The random Fourier features [2] have shown competitive performances to approximate the kernels using low-dimensional feature space. Theoretically, the more hidden features (or a larger $h$) we use, the better approximated performance the random feature should have.
>
> As mentioned in Appendix A7.5, we conducted the hyperparameter study on varying $h$. The results show that: while $h$ increases, the performance of our method on the directed graph is getting better. Even though when $h$ is as small as 5, the performance is still good where the $F_1$ score is above 70%.  Furthermore, as we can see from Figure 3 and Figure A3 where $h$ is set to 5, the $F_1$ scores of our method in general outperform other baselines in various settings.
>
> **Q4**:  "I certainly could be missing something, but the proofs provided appear to rely on large sample properties, but it's really unclear to me when those start to kick in, e.g. when should we expect the test to converge?"
>
> **A4**:  Thanks for asking this question. Actually, the large sample properties start to be needed when deriving the asymptotic distribution of the test statistic $\hat{T}$ in Theorem 4 (federated conditional independence test). Specifically, Equation (31) in Appendix A4.2 requires the assumption of large sample size, where we expect $\hat{T}$ to converge in probability to $\frac{1}{n^2} \sum_{i,j=1}^{L} \lambda_{{\ddot{X}|Z},i} \lambda_{Y|Z,j} \alpha_{ij}^2$ as $L = n\rightarrow \infty$. Similarly, we need the large sample properties in Theorem 14 (federated unconditional independence test). We have included the discussions at the end of Section 2 of our updated manuscript.

---

> > ### Author Response · Authors · 2023-11-17
> > **Responses to Reviewer j4X7 (2/2)**
> >
> > **Q5**: "How many samples before the summary statistics applied to the random Fourier features become a reliable representation of the underlying distribution for a given source?"
> >
> > **A5**: Thanks for this insightful question. Based on our empirical study as shown in Figure 3 and Figure A3, at least 100 samples for each client can provide a reliable representation of the underlying distribution for a given source.
> >
> > **Q6**: "In general, it would be good to get a sense of the finite sample behavior of the proposed method."
> >
> > **A6**: Thank you so much for the constructive suggestions. We totally agree with you. Although we need large sample properties in the proof of Theorem 4, in practice we only have finite samples. According to the experiment of varying samples in Figure 3 and Figure 4, we can see that with more samples the performance of our method is getting better. In light of your suggestion, we have included this discussion in Section 4 of our updated manuscript.
> >
> >
> > We sincerely thank you once again for the constructive and insightful suggestions. We hope you will find that our responses, along with updated manuscripts and new experiments, have properly addressed your concerns. Please kindly let us know if there are any further questions or comments.
> >
> >
> > ***
> > **References:**
> >
> > [1] Zhang, Peters, et al. "Kernel-based conditional independence test and application in causal discovery." UAI, 2012.
> >
> > [2] Rahimi and Recht. "Random features for large-scale kernel machines." NeurIPS, 2007.

---

> ### Author Response · Authors · 2023-11-22
> **Could you please kindly check whether our responses properly addressed your concerns?**
>
> Dear Reviewer j4X7:
>
> Thank you for your time and efforts in reviewing our paper. We have provided responses and revised the paper according to your constructive comments.
>
> Could you please kindly check whether our responses properly addressed your concerns? If any explanations remain unclear, we will gladly provide further clarification.
>
> Sincerely,
> Authors

---

### Official Review · Reviewer_71ck · 2023-11-08

**Soundness:** 4 excellent
**Presentation:** 4 excellent
**Contribution:** 4 excellent
**Rating:** 8
**Confidence:** 3

**Summary:**

The authors propose a novel constraint-based federated causal discovery method, FedCDH, specifically tailored for heterogenous data distributions mostly existent in FL. Novelty is twofold: Using summary statistics as a surrogate for skeleton discovery and introducing a surrogate variable to model distribution changes. The paper is very well written and shows promising results.

**Strengths:**

(1) Topic is very relevant causal inference and ICLR communities.

(2) Nicely written paper with lots of background material and well-documented code.

(3) Federated causal discovery from heterogeneous data is challenging in theoretical and realistic experiment settings.

(4) Ability to operate over general functional models.

**Weaknesses:**

(1) Communication cost for small sample size.

(2) Why have a maximum number of clients of 10?

**Questions:**

(1) Why the assumption " We set the sample size of each client to be equal." is required? Data heterogeneity can come in number of samples, too. Having unequal samples across clients is realistic.

(2) How the datasets are divided into clients is unclear to me.

---

> ### Author Response · Authors · 2023-11-17
> **Responses to Reviewer 71ck**
>
> We greatly appreciate the reviewer’s time, encouraging comments, and constructive suggestions. We have carefully **modified the manuscript** according to your detailed suggestions. Please find the responses to all your comments below. Q1 and Q2 below correspond to the points in “Weaknesses”, while Q3 and Q4 correspond to the points in “Questions”.
>
> **Q1**: “Communication cost for small sample size.”
>
> **A1**: We sincerely appreciate your comment. As mentioned in the last Section “Discussion and Conclusion”, we acknowledged that one of the limitations of our paper is that: the efficiency of our summary statistics in reducing communication costs may not be considerable when the sample size n is small, because the raw data is in dimension $\mathbb{R}^{n{\times}d'}$ while the constructed covariance tensor is in dimension $\mathbb{R}^{d'×d'×h×h}$. However, even though the sample size is small, using our constructed covariance tensor could still enjoy other advantages, such as the preservation of data privacy, as mentioned in Section 3.4.
>
> **Q2**: “Why have a maximum number of clients of 10?”
>
> **A2**: Thanks a lot for the question. We would like to clarify that: following previous work [1], the number of clients in our synthetic experiments include 2, 4, 8, 16, and 32; therefore, the maximum number of clients we considered is 32 instead of 10. Moreover, our method actually does not require constraint on the number of clients. In other words, the number of clients can be any positive number.
>
>
> **Q3**: “Why the assumption ‘We set the sample size of each client to be equal’ is required? Data heterogeneity can come in number of samples, too. Having unequal samples across clients is realistic.”
>
> **A3**: Thank you so much for this insightful point. We would like to clarify that: following the setting of previous works such as [1, 2], we therefore set the sample size of each client to be equal. However, this setting is **not a required assumption** in our method. Actually, our method can handle both equal and unequal samples across clients. As we mentioned in the paper (the paragraph above Theorem 8), our constructed global covariance tensor $\mathcal{C_T}$ is obtained by simply adding up the local ones $\mathcal{C_{T_k}}$, i.e., $\mathcal{C_T}=\sum_{k=1}^K \mathcal{C_{T_k}}$, which is not related to the local sample size here. Therefore, our method can work well no matter whether each client has an equal or unequal sample size. We have added those details to our manuscript.
>
>
> **Q4**: “How the datasets are divided into clients is unclear to me.”
>
> **A4**: Thanks for the great question. In our experiments, regarding the generation of synthetic heterogeneous data, we separately generate the data for each domain with different causal models and then combine them together, instead of generating the data and subsequently dividing. We have included this detail in our manuscript.
>
>
> Thank you again for all these constructive comments, which are really helpful in improving the quality of the manuscript. We hope our responses and the modified manuscript can adequately address the concerns. Please let us know if there are any further questions.
>
>
> ***
> **References:**
>
> [1] Ng and Zhang. "Towards federated bayesian network structure learning with continuous optimization." AISTATS, 2022.
>
> [2] Gao, Chen, et al. "FedDAG: Federated dag structure learning." TMLR, 2022.

---

> ### Author Response · Authors · 2023-11-22
> **Could you please kindly check whether our responses properly addressed your concerns?**
>
> Dear Reviewer 71ck:
>
> We sincerely appreciate your valuable time, constructive suggestions, and encouragement. Given that the rolling discussion period is coming to an end, could you please kindly provide your feedback on our responses? If there is any other concern, could you please kindly let us know? Thank you very much!
>
> Best Regards,
>
> Authors

---

> > ### Comment · Reviewer_71ck · 2023-11-23
> > **Response to authors**
> >
> > Dear authors,
> >
> > Thanks for your contributions and for your detailed response. You have answered my concerns and I maintain my positive score.

---

> > > ### Author Response · Authors · 2023-11-23
> > > **Thank you so much for checking the response and your encouragement**
> > >
> > > Dear Reviewer 71ck:
> > >
> > > We are glad that our response has well addressed your concerns. Thank you again for your valuable time, constructive suggestions, and encouragement!
> > >
> > > Sincerely,
> > > Authors

---

### Author Response · Authors · 2023-11-17
**Responses to All Reviewers**

Thanks a lot for the constructive reviews from all the reviewers. We are encouraged by some of your positive comments and also inspired by your insightful suggestions which really help in improving the quality of our manuscript. We here provide more details regarding where and how we have modified our manuscript.

- To All Reviewers:
    - More details have been included in Section 2, Section 3, and Section 4.
    - To improve the readability, we have added a symbol table in Appendix A1.
    - More experimental results and analysis are provided in Appendix A7 and A8.
    - We have moved the previous Figure 4 to Appendix Figure A3 to save more space.

- To Reviewer j4X7:
     - We have added new experiments to evaluate the power of the test, as shown in Appendix A7.8.

- To Reviewer VxyA:
     - We have added new experiments to evaluate the impact on the order of domain indices, as shown in Appendix A7.9.

---

### Meta-Review · Area_Chair_PoQK · 2023-12-06

**Metareview:**

This paper received mixed reviews. On one side, a subset of reviewers found the use of KCIT and random Fourier features for federated causal discovery clever, and thought the presentation was clear. Another subset found the presentation ambiguous. One reviewer felt that the tradeoff of number of samples vs. test powers compared to non-federated version of the proposed algorithm was not sufficiently discussed in the manuscript, which I think is a very good comment that the authors should address.

**Justification For Why Not Higher Score:**

The paper extends the existing work in federated causal discovery to non-parametric settings but I agree with Reviewer j4X7 that the tradeoffs should have been discussed in more depth since this is the main claimed contribution of the paper.

**Justification For Why Not Lower Score:**

The reviewers who recommend reject were mostly concerned about presentation clarity which I believe is relatively easy to fix in camera ready.

---

### Decision · Program_Chairs · 2024-01-16

Accept (poster)